# Beta 1-integrin–c-Met cooperation reveals an inside-in survival signalling on autophagy-related endomembranes

Rachel Barrow-McGee[1],[*], Naoki Kishi[1],[*], Carine Joffre[1],[*],[†], Ludovic Ménard[1],[*],[†], Alexia Hervieu[1],[†], Bakhouche A. Bakhouche[1], Alejandro J. Noval[1], Anja Mai[2], Camilo Guzmán[2], Luisa Robert-Masson[1],[†], Xavier Iturrioz[3],[†], James Hulit[1],[†], Caroline H. Brennan[4], Ian R. Hart[5], Peter J. Parker[3],[6], Johanna Ivaska[2],[7] & Stéphanie Kermorgant[1]

Receptor tyrosine kinases (RTKs) and integrins cooperate to stimulate cell migration and tumour metastasis. Here we report that an integrin influences signalling of an RTK, c-Met, from inside the cell, to promote anchorage-independent cell survival. Thus, c-Met and β1-integrin co-internalize and become progressively recruited on LC3B-positive 'autophagy-related endomembranes' (ARE). In cells growing in suspension, β1-integrin promotes sustained c-Met-dependent ERK1/2 phosphorylation on ARE. This signalling is dependent on ATG5 and Beclin1 but not on ATG13, suggesting ARE belong to a non-canonical autophagy pathway. This β1-integrin-dependent c-Met-sustained signalling on ARE supports anchorage-independent cell survival and growth, tumorigenesis, invasion and lung colonization in vivo. RTK–integrin cooperation has been assumed to occur at the plasma membrane requiring integrin 'inside-out' or 'outside-in' signalling. Our results report a novel mode of integrin–RTK cooperation, which we term 'inside-in signalling'. Targeting integrin signalling in addition to adhesion may have relevance for cancer therapy.

[1] Spatial Signalling Team, Centre for Tumour Biology, Barts Cancer Institute—A Cancer Research UK Centre of Excellence, Queen Mary University of London, John Vane Science Centre, Charterhouse Square, London EC1M 6BQ, UK. [2] University of Turku, Centre for Biotechnology and VTT Technical Research Centre of Finland, FI-20520 Turku, Finland. [3] Protein Phosphorylation Laboratory, Francis Crick Institute, 44 Lincoln's Inn Fields, London WC2A 3PX, UK. [4] School of Biological and Chemical Sciences, Queen Mary University of London, 327 Mile End Road, London E1 4NS, UK. [5] Centre for Tumour Biology, Barts Cancer Institute—A Cancer Research UK Centre of Excellence, Queen Mary University of London, John Vane Science Centre, Charterhouse Square, London EC1M 6BQ, UK. [6] Division of Cancer Studies, King's College School of Medicine, St Thomas Street, London SE1 1UL, UK. [7] Department of Biochemistry and Food Chemistry, University of Turku, FI-20520 Turku, Finland. * These authors contributed equally to this work. † Present addresses: Cancer Research Center of Toulouse, UMR1037, 31037 Toulouse, France (C.J.); Astrazeneca, Oncology iMed, CR-UK-Cambridge Institute, Li Ka Shing Centre, Robinson Way, Cambridge CB2 0RE, UK (L.M.); Cancer Research UK Cancer Therapeutics Unit, Haddow Laboratories, The Institute of Cancer Research, 15 Cotswold Road, Belmont, Sutton, Surrey,SM2 5NG, UK (A.H.); University College London, Infection and Immunity, Cruciform Building, 90 Gower Street, London WC1E 6BT, UK (L.R.-M.); Centre Interdisciplinaire de Recherche en Biologie (CIRB), CNRS-UMR 7241, INSERM 1050, Collège de France, 11 place Marcelin Berthelot, 75231 Paris, France (X.I.); Novintum Bioscience, London Bioscience Innovation Centre, 2 Royal College Street, London NW1 0NH, UK (J.H.). Correspondence and requests for materials should be addressed to S.K. (email: s.kermorgant@qmul.ac.uk).

C-Met, overexpressed or mutated in cancer, represents a major therapeutic target[1,2]. Binding to its ligand (hepatocyte growth factor (HGF)), triggers cell proliferation, survival and migration[1,2]. c-Met signalling post-internalization[3–5] is required for cell migration, tumour growth and metastasis[3,6]. Thus, c-Met mutations in the kinase domain are oncogenic not only because they activate c-Met, but also because they promote signalling from endosomes[6]. However, mechanisms regulating c-Met/RTK (receptor tyrosine kinase) signalling post-endocytosis, are poorly understood.

Integrins, extracellular matrix transmembrane receptors, also control tumour cell migration/invasion proliferation and survival[7–9] via bi-directional signalling. Ligand binding in the extracellular matrix (ECM), induces integrin 'outside-in signalling'; involving receptor clustering and activation, evoking intracellular signalling and cellular responses. In contrast, signals from other receptors, including RTKs, can trigger 'inside-out-signalling', where intracellular proteins interacting with the cytoplasmic face of integrins alter their activity, increasing affinity towards the matrix[7,10]. Integrin trafficking, involving constant plasma membrane-endosomes shuttling facilitating the dynamic regulation of cell adhesion, plays a vital role in regulating cell migration[10–13].

Integrin–RTK cooperation plays a major role in cellular outcome[7]. However, mechanisms, especially how the cooperation is spatially orchestrated, are poorly defined. Integrins can bind RTKs directly, promoting their activation[14–16] or internalization[17]. Conversely, RTKs can increase integrin expression[18,19], activation[20] and recycling[21]. However, there is no evidence that integrins and RTKs can cooperate on endomembranes.

Here we show, in several models including breast and lung cancer cells, that β1-integrin positively regulates the endocytosis of activated c-Met as well as c-Met signalling post-endocytosis, unexpectedly from autophagy-related endomembranes ('ARE'), likely part of a non-canonical autophagy pathway. This β1-integrin cooperation, occurs in cells grown in suspension and leads to increased anchorage-independent cell survival/ growth. We report a novel mode of integrin–RTK cooperation, which we term 'inside-in signalling'.

## Results

**c-Met and β1-integrin co-internalize in a molecular complex.** β1-Integrin is the β-subunit of most ECM binding integrins, including α5β1 the major fibronectin receptor[7–9]. To investigate whether the c-Met pathway influences the trafficking of β1-integrin, Flow cytometry, biotin internalization assays and confocal imaging were used in the following cell models: 'β1A cells' corresponding to β1-integrin null GD25 cells[22] re-expressing β1-integrin[23]; NIH3T3 cells expressing c-Met mutant M1268T, the oncogenicity of which results from constitutive activation and endocytosis/trafficking[6]; and the human epithelial, breast MDA-MB-468 and non-small cell lung carcinoma A549 cell lines[24,25]. Either c-Met activation upon HGF stimulation or constitutive activation triggered internalization of a pool of surface β1-integrin (Fig. 1a,b,d,e, Supplementary Fig. 1a–d). Interestingly, colocalization between internalized c-Met/fluorescently labelled HGF (HGF-AlexaFluor-555)[4] and β1-integrin was observed (Fig. 1d,e; Supplementary Fig. 1d; Supplementary Data 1). Moreover, live confocal imaging, using an anti-β1-integrin antibody conjugated to AlexaFluor-488 or integrin-α5-GFP, together with HGF-AlexaFluor-555, demonstrated that the two molecules co-internalize and co-traffic (Supplementary Movies 1 and 2).

We then analysed 'c-Met-GFP cells', which are HEK-293 cells with tetracycline-inducible expression ('TET on') of constitutively phosphorylated c-Met-GFP[26] (Supplementary Fig. 1e) which induces cell detachment, leading to floating, viable colonies (Supplementary Fig. 1f; Supplementary Movies 3 and 4) that retain c-Met-GFP kinase activity. This induced cell detachment coincided with β1-integrin internalization (Fig. 1c); while total cellular β1-integrin levels remained unaltered (Supplementary Fig. 1e). Immunofluorescence (on cells seeded on poly-L-lysine-coated coverslips) showed that c-Met and β1-integrin colocalized in intracellular vesicles (Fig. 1f) while live imaging demonstrated a constitutive trafficking of c-Met-GFP (Supplementary Movies 5). c-Met and β1-integrin also co-internalize in A549 and MDA-MB-468 cells maintained in suspension, detected post-HGF treatment for 120 min (Fig. 1g,h).

c-Met and β1-integrin association in a complex was detected by co-immunoprecipitation and proximity ligation assay (PLA) in adherent and detached cells, both without and with HGF stimulation (Fig. 1i,j, Supplementary Fig. 1g,h). β1-Integrin co-immunoprecipitated with c-Met from intracellular fractions, obtained with a 'biotin surface removal assay' (see Methods section) (Fig. 1k), confirming an intracellular association of the two molecules in a complex.

Thus, under both adherent and detached conditions, a proportion of c-Met and β1-integrin associate: (i) in a molecular complex at the plasma membrane under basal conditions; (ii) on endomembrane, following c-Met–β1-integrin co-internalization, upon c-Met activation.

**β1-Integrin promotes c-Met-sustained ERK1/2 signalling.** Previous studies showed that c-Met needs to internalize to signal[3,4,6,24,27]. Therefore we wondered whether β1-integrin can affect c-Met signalling. Strikingly, in all cell models, the absence of β1-integrin significantly impaired sustained c-Met-dependent ERK1/2 phosphorylation while c-Met expression and phosphorylation levels were unchanged. This occurred in GD25, compared to β1A cells, stimulated with HGF for up to 120 min (Fig. 2a; Supplementary Fig. 2a,b), and in cells knocked down for β1-integrin including M1268T cells (Fig. 2b; Supplementary Fig. 2c–f), c-Met-GFP in suspension 16 h post-tetracycline (Fig. 2c, Supplementary Fig. 2g–i), A549 (Fig. 2d; Supplementary Fig. 2j) and MDA-MB-468 (Supplementary Fig. 2j,k), cultured in suspension for 120 min with HGF. Thus, β1-integrin influences downstream signalling of c-Met in a manner independent of cell adhesion.

**β1-Integrin promotes c-Met-driven *in vivo* tumorigenesis.** The functional importance of β1-integrin in c-Met signalling was assessed in *in vivo* tumour growth and experimental metastasis. NIH3T3 cells expressing the c-Met oncogenic mutant M1268T rapidly formed tumours (sensitive to c-Met inhibition) in nude mice[6]. Tumour volumes and weight were reduced significantly (50–60% and 2.5-fold respectively; $t$-test $P < 0.001$ to 0.05) in β1-integrin versus control small interfering RNA (siRNA)-transfected cells (Fig. 2e; Supplementary Fig. 2l); while no difference occurred in wild-type (WT) cells (Supplementary Fig. 2m).

In a lung colonization assay, we have previously shown that only NIH3T3 cells expressing the c-Met oncogenic mutant M1268T but not c-Met WT colonized the lung[6]. At 21 days, mice injected with control siRNA-transfected M1268T cells presented an average of 8.5 macroscopic pulmonary tumours compared with 2.8 in mice injected with β1-integrin knocked down cells ($t$-test, $P < 0.05$) (Fig. 2f). Most lung tissue was invaded by control siRNA-transfected cells; lung tissue of mice injected with β1-integrin siRNA-transfected cells lacked cell invasion (Supplementary Fig. 2n).

In a 24 h *in vivo* invasion assay in zebrafish embryos, M1268T cells were more invasive than WT cells, with the invasion of M1268T cells inhibited by the c-Met inhibitor PHA-665752 (Supplementary Fig. 2o). β1-Integrin siRNA knockdown significantly reduced invasion of mutant, but not WT cells (Fig. 2g).

Thus, β1-integrin is required for oncogenic c-Met-dependent *in vivo* tumour growth and invasion. Our results further suggest that β1-integrin is required for c-Met-dependent experimental lung colonization.

**β1-Integrin role in c-Met signalling is adhesion independent.** β1A and A549 cells were harvested and plated on laminin, fibronectin or poly-L-lysine, for different periods +/− HGF.

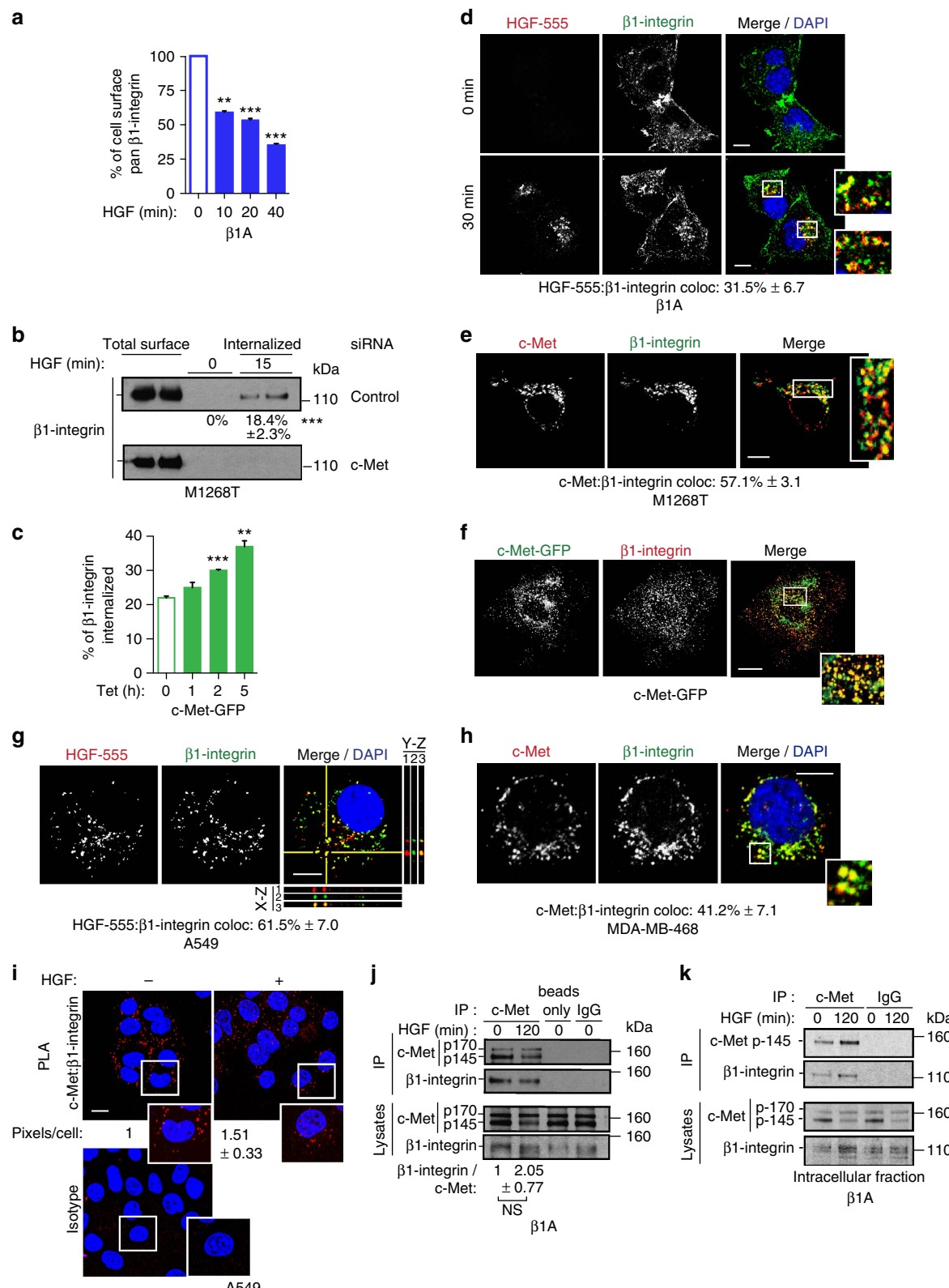

HGF activated ERK1/2 comparably under each condition (Supplementary Fig. 3a,b), suggesting that β1-c-Met-dependent ERK1/2 activation was unrelated to substrate engagement. The β1-integrin function blocking antibody, AIIB2, impaired cell adhesion (Supplementary Fig. 3c), but had no influence on HGF-stimulated ERK1/2 activation in A549 cells in suspension (Supplementary Fig. 3d).

However, c-Met was found to colocalize on endomembrane with β1-integrin in a primed conformation for ligand binding (detected with 9EG7 antibody) termed here 'active conformation' as shown in A549 cells (Fig. 3a; Supplementary Data 1). In c-Met-GFP cells treated with tetracycline for 16 h (cells totally detached), a stronger reduction (almost 60% $P < 0.001$) in cell surface levels of active conformation β1-integrin versus pan-β1-integrin (18% $P < 0.01$) was detected (Supplementary Fig. 3e,f). Reduction of active conformation β1 relative to pan β1 at the cell surface was 45% ($P < 0.01$) (Fig. 3b). This was partially restored upon pharmacological c-Met inhibition with SU11274 for 15 min ($t$-test, $P < 0.05$) Fig. 3b. The analysis of localization of active conformation β1-integrin in 293-HEK non-transfected and transfected with c-Met-GFP ('c-Met-GFP cells') seeded together, followed by 16 h tetracycline treatment, gave similar results. In non-transfected cells, β1-integrin was distributed at the plasma membrane and in intracellular pools. In c-Met-GFP cells, β1-integrin staining was increased in intracellular pools (Fig. 3c). Treatment with c-Met inhibitor SU11274 restored active conformation β1-integrin localization at the plasma membrane (Fig. 3b,d). Thus c-Met-dependent internalized β1-integrin pool is enriched in its active conformation. This process is dependent on c-Met activity and is highly dynamic. This is consistent with the finding that active conformation integrins are endocytosed more efficiently, being found on endosomes to a greater extent than pan integrins[28].

The above suggested that the active β1-integrin conformer plays a role in c-Met signalling. WT or cpdm MEFs, which are null for SHARPIN (endogenous inhibitor of β1-integrin activity)[29], were stimulated with HGF for up to 120 min, whilst in suspension. Although ERK1/2 phosphorylation was transient in WT cells, the signal was sustained in cpdm cells (Fig. 3e). Intracellular colocalization between active conformation β1-integrin and c-Met was observed at 120 min of HGF stimulation in cpdm MEFs (Supplementary Fig. 3g). PI3K inhibition, using LY294002, did not increase P-ERK1/2 in the WT MEFs at 120 min HGF stimulation, excluding the role of SHARPIN as a negative regulator of PTEN[30], in sustaining c-Met signalling in cpdm cells (Supplementary Fig. 3i). Increasing β1-integrin activity through incubating the WT cells with 1 mM MnCl$_2$ increased basal ERK1/2 activation as expected. However, a significant fold increase in ERK1/2 phosphorylation occurred upon HGF stimulation for 120 min to the same level as that observed in cpdm cells ($t$-test, $P < 0.05$, Fig. 3e and Supplementary Fig. 3h). Thus, β1-integrin in its active conformation plays a role in β1-integrin–c-Met cooperation. Moreover, c-Met activation increased intracellular β1-integrin activation levels in cells in suspension (see A549 cells treated with HGF for 120 min) (Supplementary Fig. 3j).

Thus, in detached cells, the input of β1-integrin in c-Met signalling is β1-integrin ligand-independent. However, activated c-Met increases the level of endomembrane-associated active conformation β1-integrin, which in turns positively regulates c-Met signalling.

**c-Met and β1-integrin cooperation is endocytosis dependent.** Our results suggested that the cooperation for signalling occurs inside the cells. Thus, we analysed the influence of impairing the endocytic machinery on c-Met-dependent ERK1/2 phosphorylation. Dynasore, the small GTPase dynamin inhibitor or siRNA clathrin heavy chain (CHC), reduced c-Met endocytosis (Supplementary Fig. 4a,b) and c-Met-dependent ERK1/2 phosphorylation (Fig. 4a–c). These results indicated that c-Met signals from intracellular compartments. Under these conditions, β1-integrin internalization also was reduced (Supplementary Fig. 4c), further suggesting that c-Met and β1-integrin co-internalization is required for c-Met signalling to ERK1/2.

The cytoplasmic domain of β1-integrin contains two conserved NXXY motifs implicated in matrix-stimulated β1-integrin internalization[31]. In cells expressing the β1-integrin double mutant, Y783F/Y795F, in NXXY motifs ('β1A-YYFF cells'), β1-integrin internalization was reduced dramatically after HGF stimulation, compared with WT β1-integrin in β1A cells (Fig. 4d). In β1A-YYFF cells, c-Met-dependent ERK1/2 activation was impaired as in GD25 cells, with no sustained signal at 120 min (Fig. 4e), while c-Met phosphorylation was unchanged (Supplementary Fig. 4d). The reduction in HGF-dependent ERK1/2 phosphorylation was not the result of a

**Figure 1 | c-Met and β1-integrin co-internalize in a molecular complex in both adherent cells and those in suspension** (**a**) Mean percentage cell surface β1-integrin levels ± s.e.m. in β1A cells stimulated with HGF for the indicated times, assessed by flow cytometry (fluorescence intensity, arbitrary units, $n = 3$). (**b**) Western blot for β1-integrin following a biotinylation internalization assay in M1268T c-Met-expressing NIH3T3 cells transfected with control, or c-Met siRNA. Cells were incubated for 15 min at 37 °C. Numbers are mean percentages of internalization ± s.e.m. ($n = 5$). (**c**) The mean percentage of internalized β1-integrin within 30 min (obtained with a biotinylation internalization assay) in c-Met-GFP cells treated with tetracycline (Tet) for the times indicated and compared with total cell surface β1-integrin ± s.e.m. (arbitrary units, $n = 3$). (**d–h**) Confocal sections of cells stained for DAPI (blue) (**d,g,h**), c-Met or HGF-AlexaFluor-555 (HGF-555) (red) and β1-integrin (green) (**d,e,g,h**) or expressing c-Met-GFP (green) and stained for β1-integrin (red) (**f**). Colocalizations appear in yellow. Scale bar, 10 μm (**d,e,g,h**) and 20 μm (**f**). Numbers are mean percentage colocalization ± s.e.m. ($n = 3$). (**d**) β1A cells stimulated with HGF-555 for 0 or 30 min. (**e**) M1268T c-Met-expressing NIH3T3 cells. (**f**) c-Met-GFP cells treated with tetracycline for 5 h. (**g,h**) Cells in suspension stimulated with HGF for 120 min and cytospun. (**g**) A549. Orthogonal reconstructions of 10 serial confocal slices are shown (y–z and x–z axis with 1: HGF-555, 2: β1-integrin, 3: merge of 1 and 2) alongside the one z-slice taken in the middle of the cells. The perpendicular yellow lines on the section indicate from where the orthogonal views were built. (**h**) MDA-MB-468. (**i**) proximity ligation assay (PLA). Confocal sections of A549 cells $- / +$ HGF (100 ng ml$^{-1}$) for 120 min, fixed and stained with c-Met and β1-integrin or equivalent isotyped IgG, followed by the binding of PLA probes. The red dots indicate proximity between c-Met and β1-integrin. Numbers represent the mean fold change in PLA signal (c-Met-β1-integrin) per cell normalized on total c-Met levels ± s.e.m. ($n = 2$). Scale bars, 10 μm. (**j**) Western blots for c-Met and β1-integrin following immunoprecipitation with c-Met B2 antibody, IgG control or no antibody (beads only) in β1A cells. Cells were stimulated with HGF for 0 or 120 min. Total c-Met and β1-integrin levels in the cell lysates are shown. Numbers ± s.e.m. ($n = 3$) represent the levels of β1-integrin co-immunoprecipitated, normalized to c-Met immunoprecipitate, at 0 min (levels set as 1) and 120 min of HGF stimulation (levels expressed as a fold change from 0 min). Values, obtained by densitometric analysis, were first tresholded on IgG values. (**k**) c-Met-β1-integrin co-immunoprecipitation. Following HGF stimulation of β1A cells for 0 or 120 min, cell surface proteins were biotinylated at 4 °C and removed using streptavidin pull-down. Immunoprecipitation was performed with c-Met (B2) antibody or IgG control on the intracellular fractions. Western blots for c-Met and β1-integrin post-immunoprecipitation from intracellular fractions and in the initial cell lysates are shown. $t$-Test, **$P < 0.01$; ***$P < 0.001$.

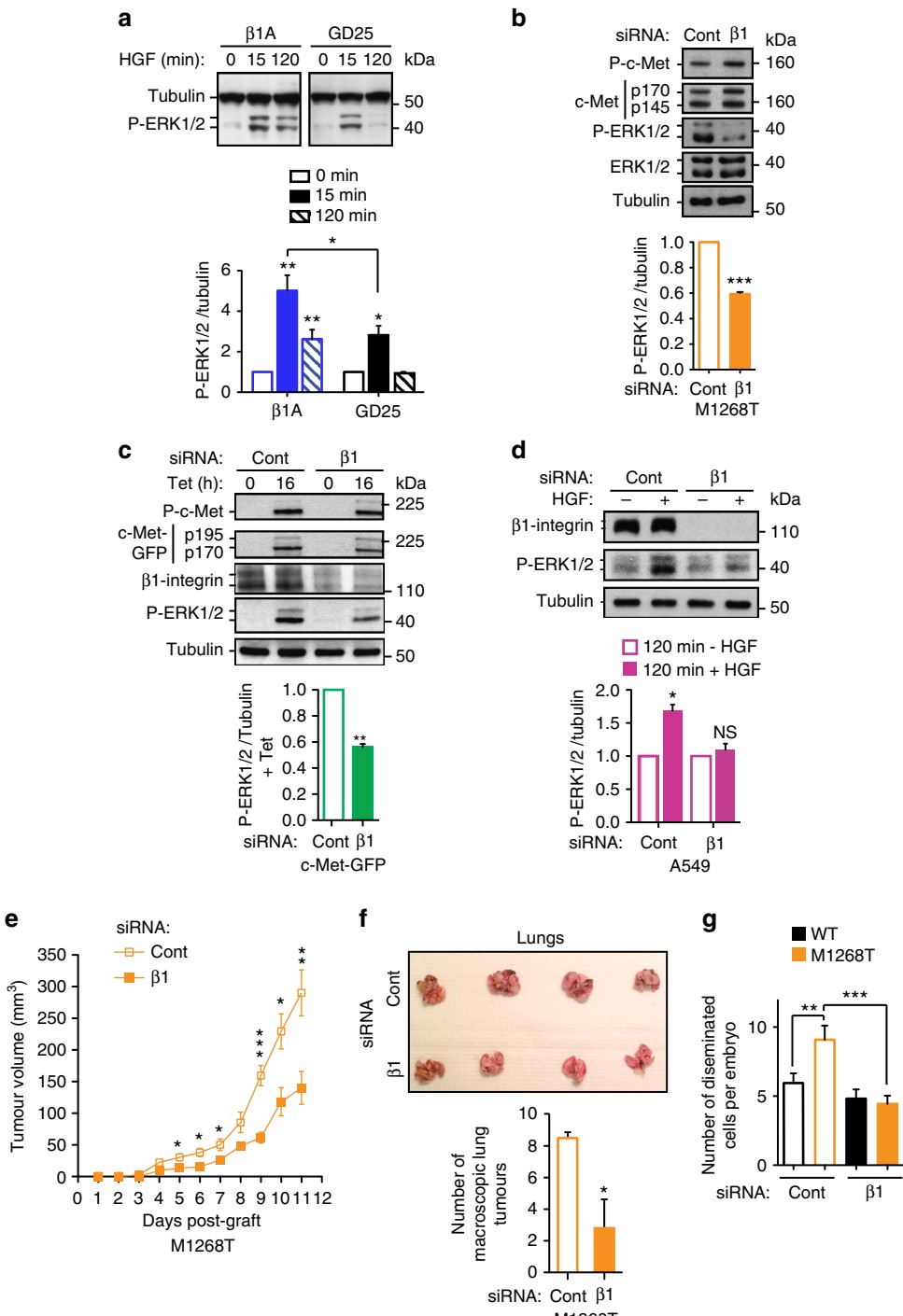

**Figure 2 | β1-integrin is required for sustained c-Met-dependent ERK1/2 phosphorylation in detached cells, c-Met-dependent *in vivo* tumorigenesis and invasion.** (**a–d**) Western blots for: (**a**) tubulin and phospho-ERK1/2 in β1A and GD25 (β1−/−) cells, stimulated with HGF for 0, 15 and 120 min; (**b**) Phospho-c-Met (Y1234-355), c-Met, phospho-ERK1/2, ERK 1/2 and tubulin in M1268T c-Met-expressing NIH3T3; (**c**) phospho-c-Met (Y1234-355), GFP (c-Met-GFP: p195, precursor; p170, mature β chain), β1-integrin, phospho-ERK1/2 and tubulin in c-Met-GFP cells incubated with tetracycline (Tet) for 0 or 16 h; (**d**) β1-integrin, phospho-ERK1/2, and tubulin in A549 cells, stimulated without (−) or with (+) HGF for 120 min in suspension; (**b–d**) All cells were transfected with control (Cont) or β1-integrin (β1) (human cells: oligo 1, Qiagen; mouse cells: oligo 3, Dharmacon) siRNA. Graphs represent phospho-ERK1/2/tubulin ratios (means ± s.e.m.), normalized to appropriate controls: (**a,d**) no HGF; (**b,c**) siRNA control (Cont), obtained by densitometric analysis (n = 3 to 6). (**e**) Tumour growth curves, over time, of M1268T c-Met-expressing NIH3T3 cells, transfected with control (Cont) or β1-integrin (β1) siRNA. Data are mean tumour volume (mm³) ± s.e.m. of n = 5 mice per group. (**f**) Pictures of the lungs of mice dissected 21 days after injection into the tail vein with either control (Cont) (n = 4) or β1-integrin (β1) (n = 5) siRNA-transfected M1268T c-Met-expressing NIH3T3 cells. Graph represents the mean number of macroscopic tumours per mouse ± s.e.m. (**g**) Mean number ± s.e.m. of disseminated WT and M1268T c-Met-expressing NIH3T3 cells per zebrafish embryo 24 h after injection. Cells were transfected with control (Cont) or β1-integrin (β1) siRNA (n = 3, average of 25 embryos per condition per experiment). *t*-Test, * $P < 0.05$; **$P < 0.01$; ***$P < 0.001$.

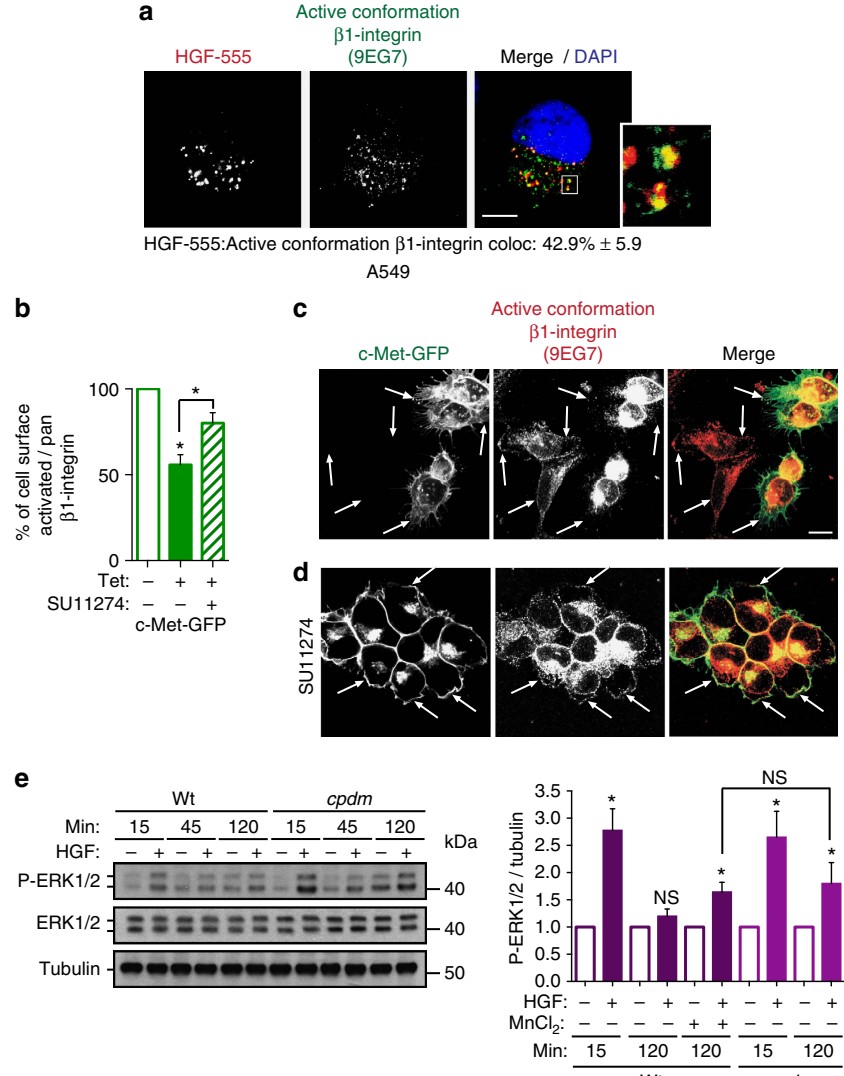

**Figure 3 | The role of β1-integrin in c-Met-dependent signalling is adhesion independent though its active conformation is a positive regulator.**
(**a**) Confocal section of A549 cells stimulated with HGF-AlexaFluor-555 (HGF-555, red) for 120 min in suspension. Cells were cytospun, fixed and stained for active conformation β1-integrin (9EG7, green) and DAPI (blue). Colocalizations appear in yellow. Scale bar, 10 μm. Numbers are mean percentage colocalization ± s.e.m. (n = 3). (**b**) The mean percentage cell surface levels ± s.e.m. of active conformation β1-integrin (9EG7) reported on pan-β1-integrin (DF7) assessed by flow cytometry. c-Met-GFP cells were treated with or without tetracycline (Tet) for 16 h (arbitrary units, n = 4) and with or without SU11274 (2 μM) (n = 3). (**c,d**) Confocal projections of 7 Z-sections from the base to the apex of cells. Arrows show examples of plasma membrane staining. Scale bar, 10 μm. Cells were cultured on Poly-L-lysine coated glass coverslips for 16 h with tetracycline and stained for active conformation β1-integrin (9EG7, red). c-Met-GFP is in green. (**c**) T-REx-293 cells, non-transfected and stably transfected with c-Met-GFP ('c-Met-GFP cells'), at a 50/50 ratio. (**d**) c-Met-GFP cells in the presence of the c-Met inhibitor SU11274 (2 μM). (**e**) Western blots for phospho-ERK1/2, ERK1/2 and tubulin in WT and *cpdm* (SHARPIN null) MEFs, stimulated without ( − ) or with ( + ) HGF for 15 and 120 min in suspension and treated without ( − ) or with ( + ) 1 mM MnCl₂. Graphs represent phospho-ERK1/2/ERK1/2 ratios ± s.e.m. Normalized to no HGF obtained by densitometric analysis (n = 3). t-Test, *P < 0.05.

decrease in ERK1/2 expression levels in GD25/β1A-YYFF, compared with β1A cells (Supplementary Fig. 4e). Thus, the effects of β1-integrin on c-Met signalling depend on a trafficking-competent β1-integrin with intact cytoplasmic NXXY motifs.

**β1-Integrin is required for c-Met endocytosis.** The HGF-dependent rate of c-Met internalization also was reduced markedly in β1A-YYFF versus β1A cells (Fig. 4f,g), suggesting that β1-integrin internalization is required for optimal HGF-mediated c-Met internalization. Accordingly c-Met internalization was reduced in cells lacking β1-integrin, including GD25 compared with β1A cells (Fig. 4g; Supplementary Fig. 4f), A549 and MDA-MB-468 cells knocked down for β1-integrin and grown in

suspension (Fig. 4h; Supplementary Fig. 4g). Conversely, c-Met internalization was increased significantly in *cpdm* cells (Fig. 4i). Thus, active conformation β1-integrin not only co-internalizes with activated c-Met but also is required for optimal c-Met internalization.

Since endocytosis is required for optimal c-Met signalling, we hypothesized that the role of β1-integrin in c-Met signalling is a consequence of its role on c-Met endocytosis. We thus reasoned that rescuing c-Met internalization in cells expressing a β1-integrin form defective in internalization, such as β1A-YYFF, would restore signalling. Rab21 promotes β1-integrin endocytosis[32]. β1A-YYFF cells expressed lower levels of Rab21 compared with β1A cells (Supplementary Fig. 4h). The expression of GFP-Rab21 in β1A-YYFF cells restored HGF-AlexaFluor-555 uptake

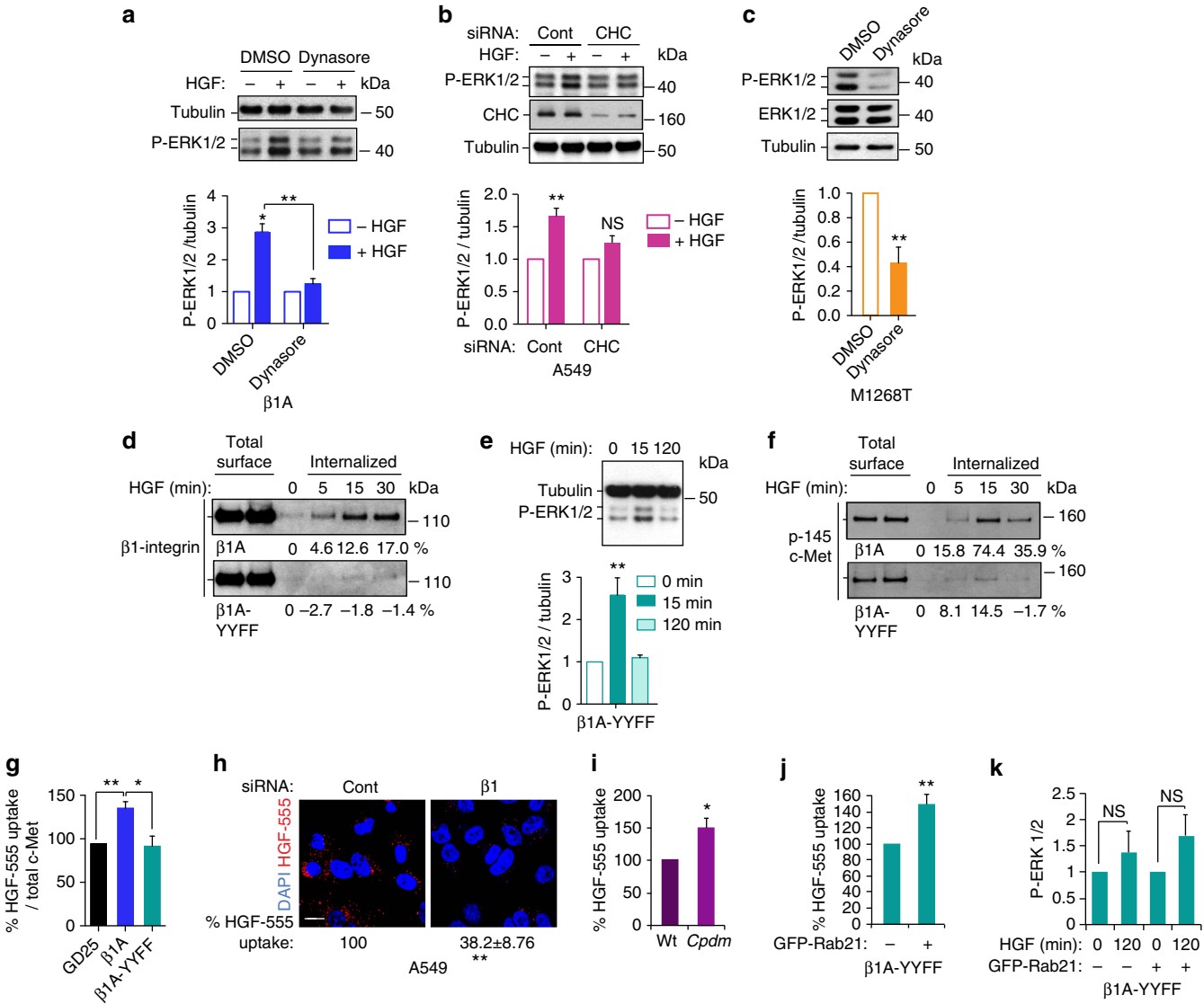

**Figure 4 | c-Met and β1-integrin cooperation is endocytosis dependent and β1-integrin is required for c-Met endocytosis. (a)** Western blots for phospho-ERK1/2, ERK1/2 and tubulin in β1A cells stimulated without ( − ) or with ( + ) HGF for 120 min following pre-treatment with DMSO or Dynasore (80 μM); **(b)** western blots for phospho-ERK1/2, clathrin and tubulin in A549 cells transfected with control (Cont) or clathrin heavy chain (CHC) siRNA and incubated in suspension for 120 min without ( − ) or with ( + ) HGF. **(c)** Western blots for phospho-ERK1/2, ERK1/2 and tubulin in M1268T c-Met-expressing NIH3T3 treated with DMSO or Dynasore (80 μM). **(a–c)** Graphs represent mean phospho-ERK1/2/tubulin ratio ± s.e.m.; **(a,b)** upon HGF stimulation normalized to no HGF; **(c)** with dynasore normalized to DMSO, obtained by densitometric analysis ($n = 3$). **(d)** Western blot for β1-integrin following a biotinylation internalization assay in β1A and β1A-YYFF cells incubated for 0, 5, 15 and 30 min at 37 °C with HGF. Numbers represent the percentage of internalization. **(e)** Western blots for tubulin and phospho-ERK1/2 in β1A-YYFF cells, stimulated with HGF for 0, 15 and 120 min. Graph is mean phospho-ERK1/2/tubulin ratio ± s.e.m. Normalized to HGF 0 min, obtained by densitometric analysis ($n = 6$). **(f)** Western blot for c-Met following a biotinylation internalization assay in β1A and β1A-YYFF cells incubated for 0, 5, 15 and 30 min at 37 °C with HGF. Numbers represent the percentage of internalization. **(g)** Percentage of HGF-AlexaFluor-555 (HGF-555) uptake (mean red pixels per cell/total c-Met levels) ± s.e.m. after 15 min incubation in β1A and β1A-YYFF cells normalized to the uptake in GD25 cells ($n = 3$). **(h)** Confocal sections of A549 cells, stimulated with HGF-AlexaFluor-555 (HGF-555) for 120 min in suspension. Scale bar, 10 μm. Numbers are percentage of HGF-555 uptake (mean red pixels per cell) ± s.e.m. in cells transfected with β1-integrin siRNA, normalized to the uptake in cells transfected with control siRNA ($n = 3$). **(i)** Percentage of HGF-AlexaFluor-555 (HGF-555) uptake (mean red pixels per cell) ± s.e.m. in cpdm (SHARPIN null) MEFs stimulated for 120 min in suspension, normalized to the uptake in WT MEFs ($n = 3$). **(j)** Percentage of HGF-AlexaFluor-555 (HGF-555) uptake (mean red pixels per cell/total c-Met levels) ± s.e.m. in β1A-YYFF cells positive for GFP-Rab21 normalized to the uptake in β1A-YYFF cells negative for GFP-Rab21 (from the same coverslips) ± s.e.m. ($n = 3$). **(k)** Mean phospho-ERK1/2 levels ± s.e.m. in permeabilised β1A-YYFF cells, Rab21-GFP negative ( − ) or positive ( + ) (from the same population), upon stimulation with HGF for 120 min, normalized to mean phospho-ERK1/2 levels at HGF 0 min, assessed by flow cytometry ($n = 3$). t-Test, *$P < 0.05$; **$P < 0.01$; NS: not significant.

to levels observed in β1A cells (Fig. 4j, compared with Fig. 4g). However HGF-dependent ERK1/2 activation was not rescued, as assessed by flow cytometry analysis of GFP-positive cells (Fig. 4k, Supplementary Fig. 4i), suggesting that β1-integrin, and its cytoplasmic NXXY domain, is not only required for optimal c-Met endocytosis but also has an additional role in c-Met signalling post-internalization.

c-Met and β1-integrin continue to co-traffic post-internalization with colocalizations detected at 120 min of HGF stimulation (Fig. 1g,h) and β1-integrin mostly influences the sustained c-Met-

dependent ERK1/2 activation (Fig. 2). Since endocytosed integrins normally return to the plasma membrane within 15–30 min the prolonged c-Met–integrin intracellular colocalizations suggested that β1-integrin might play a 'signalling' function from an intracellular compartment not previously associated with integrin traffic.

**c-Met and β1-integrin co-traffic on LC3B-positive vesicles.** We investigated, initially using adherent cells, where c-Met and β1-integrin co-traffic following HGF stimulation, through monitoring colocalization with EEA1 (early endosome antigen 1), Rab4-GFP (early recycling), Rab11-GFP (late recycling),

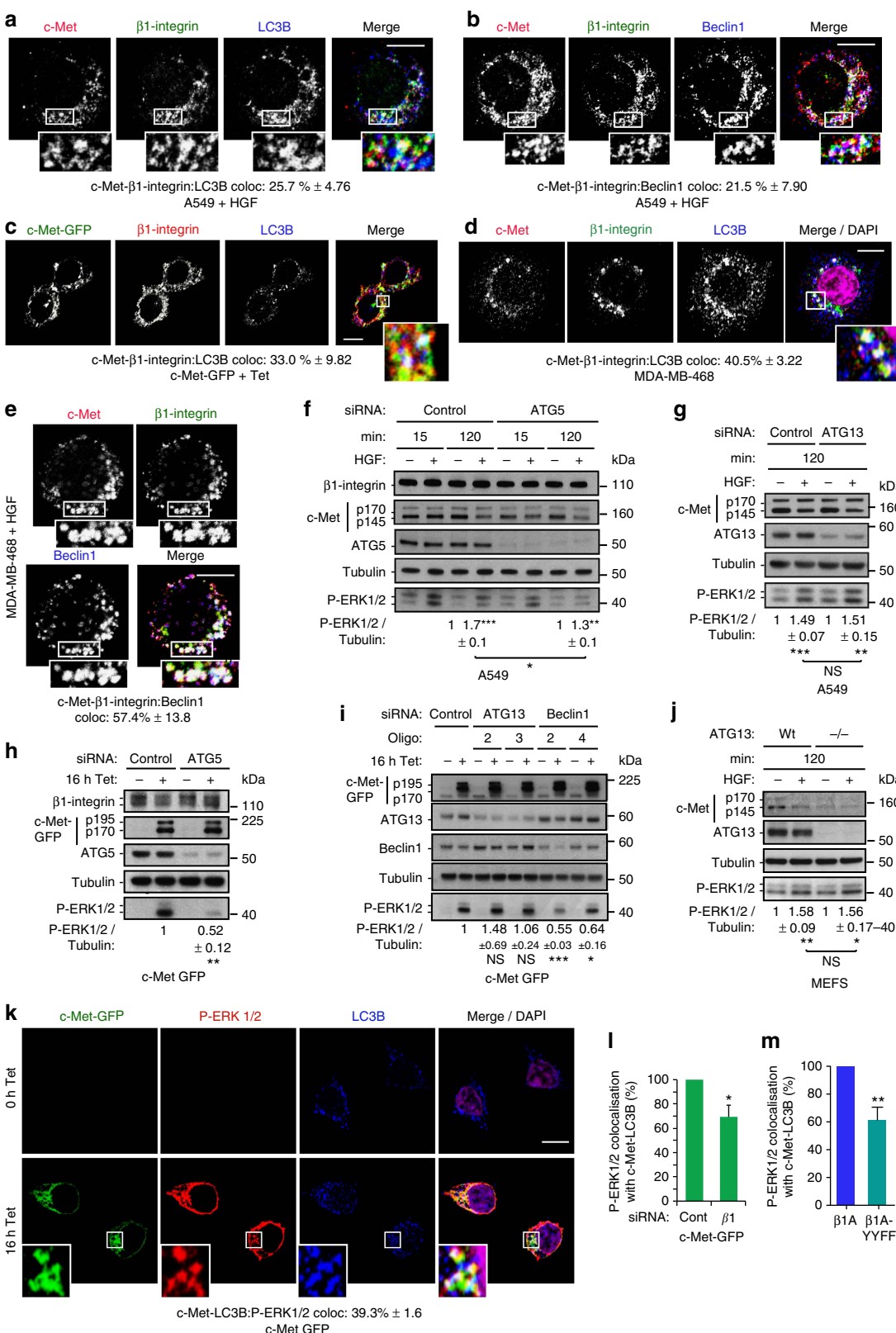

**a** c-Met β1-integrin LC3B Merge
c-Met-β1-integrin:LC3B coloc: 25.7 % ± 4.76
A549 + HGF

**b** c-Met β1-integrin Beclin1 Merge
c-Met-β1-integrin:Beclin1 coloc: 21.5 % ± 7.90
A549 + HGF

**c** c-Met-GFP β1-integrin LC3B Merge
c-Met-β1-integrin:LC3B coloc: 33.0 % ± 9.82
c-Met-GFP + Tet

**d** c-Met β1-integrin LC3B Merge / DAPI
c-Met-β1-integrin:LC3B coloc: 40.5% ± 3.22
MDA-MB-468

**e** c-Met β1-integrin Beclin1 Merge
MDA-MB-468 + HGF
c-Met-β1-integrin:Beclin1 coloc: 57.4% ± 13.8

**k** c-Met-GFP P-ERK 1/2 LC3B Merge / DAPI
0 h Tet / 16 h Tet
c-Met-LC3B:P-ERK1/2 coloc: 39.3% ± 1.6
c-Met GFP

Rab7-GFP (late endosome) and LC3B (ref. 33) (autophagosomes and LC3B-associated phagocytosis).

Although some overlap with each marker was detected, at late time points (especially 120 min) the greatest enrichment of c-Met–β1-integrin occurred with LC3B in MDA-MB-468 and A549 cells (Supplementary Fig. 5a–d). Triple colocalizations also were observed in β1A cells at 120 min HGF stimulation (13.2%, Supplementary Fig. 5e), M1268T cells (30.9%, Supplementary Fig. 5f) and in *cpdm* cells at 120 min HGF stimulation (Supplementary Fig. 5g). Thus, co-internalized c-Met and β1-integrin appear to traffic progressively to LC3B-positive compartments. Triple colocalization also occurred in cells in suspension including A549 (25.7%, Fig. 5a, Supplementary Data 1) and MDA-MB-468 (40.5%, Fig. 5d, Supplementary Data 1), at 120 min HGF stimulation, and in 16 h Tet-induced c-Met-GFP cells (33%, Fig. 5c, Supplementary Data 1). β1-Integrin–c-Met colocalizations with Beclin1, another marker of autophagosomes and LC3B-associated phagocytosis, were also observed as shown in A549 (21.5%, Fig. 5b, Supplementary Data 1) and MDA-MB-468 (57.4%, Fig. 5e, Supplementary Data 1), in suspension with HGF for 120 min.

**c-Met and β1-integrin signal on ARE**. To assess whether the localization of c-Met and β1-integrin on LC3B/Beclin1 endomembranes is important for β1-integrin–c-Met signalling, the formation of such endomembranes was reduced by knocking down the autophagy regulator ATG5 (autophagy protein 5) using siRNA, which decreased the levels of lipidated LC3B, LC3BII (Supplementary Fig. 5h). This resulted in a significant reduction of c-Met-dependent sustained ERK1/2 phosphorylation, to levels seen in β1-integrin-depleted cells, as shown in A549 and MDA-MB-468 cells in suspension stimulated with HGF for 120 min and in c-Met-GFP cells + 16 h tetracycline (Fig. 5f,h, Supplementary Fig. 5k). However c-Met phosphorylation (Supplementary Fig. 5l), expression levels (Supplementary Fig. 5m) and c-Met endocytosis (Supplementary Fig. 5n) were unaffected. In *cpdm* cells (where ERK1/2 activation was increased at 120 min HGF compared with WT cells, Fig. 3e), a decrease in ERK1/2 phosphorylation was detected at 120 min (Supplementary Fig. 5o). These results were confirmed using Beclin1 siRNA (Fig. 5i). Recently, a non-canonical autophagy pathway, called LAP for 'LC3B-associated phagocytosis' was described[34–37]. While ATG5 and Beclin1 are players involved in canonical and non-canonical autophagy, ATG13 plays a role only in the canonical autophagy[34–37]. Interestingly, ATG13 siRNA-mediated knockdown had no effect on c-Met-dependent ERK1/2 signalling in c-Met-GFP cells, post-16 h tetracycline (Fig. 5i) and A549 cells at 120 min of HGF stimulation in suspension (Fig. 5g). We verified that knockdown of ATG13 or Beclin1 reduced the levels of lipidated LC3B, LC3BII (Supplementary Fig. 5i,j). Furthermore, ATG13 knockout MEFs[38] had no altered c-Met-dependent sustained ERK1/2 signalling (Fig. 5j). These results suggest that c-Met and β1-integrin co-traffic and signal on endomembranes belonging to a non-canonical autophagy pathway instead of on the autophagosome *per se*. We have called this compartment 'ARE'.

Cells were treated with the lysosomal inhibitor chloroquine, interfering with both canonical and non-canonical autophagy (witnessed by an increase in LC3BII levels)[35]. In this condition, c-Met activation, ERK1/2 activation, c-Met and β1-integrin expression levels on basal conditions and upon c-Met activation were unchanged (Supplementary Fig. 6a–h). These results indicated that c-Met signalling as well as c-Met and β1-integrin stability are not influenced by canonical/non-canonical autophagy flux. Additionally, c-Met activation (HGF dependent or constitutive) (Supplementary Fig. 6a,d,e,g) or β1-integrin levels (Supplementary Fig. 6i,j) did not affect basal autophagy as assessed with LC3B western blots. Thus, c-Met and β1-integrin appear not to influence autophagic flux in our experiments.

Autophagosomes, and related endomembranes, have been considered to be degradative rather than signalling compartments[39,40]. Our results suggest that the c-Met-β1-integrin complex activates ERK1/2 on 'ARE'; consistent with the report that autophagy proteins regulate EGF-dependent ERK1/2 activation[41]. A pool of phosphorylated ERK1/2 was detected on 'ARE', together with c-Met, upon HGF/tetracycline treatment (Fig. 5k–m, Supplementary Data 1). Moreover, P-ERK1/2-c-Met colocalization on LC3B-positive endomembrane was reduced upon β1-integrin knockdown or mutation (β1A-YYFF cells) compared to controls (control siRNA in c-Met-GFP cells and β1A cells) (Fig. 5l,m). These results further indicated that β1-integrin impinges on c-Met signalling on ARE.

**β1 may act as an adaptor to sustain c-Met signalling on ARE.** We hypothesized that, on the ARE, β1-integrin acts as a scaffold between c-Met and Shc through the NXXY motif, previously reported to modulate signalling of β3-integrin to Shc[42]. In all cells, p52[Shc] phosphorylation, unlike p66[Shc] and p46[Shc], was

**Figure 5 | c-Met and β1-integrin co-traffic and signal on autophagy-related endomembranes.** (**a–e**) Confocal sections. Scale bar, 10 μm. Numbers are percentages colocalization ± s.e.m. (**a,b**) A549 cells at 120 min HGF stimulation in suspension, stained for c-Met (red), β1-integrin (green) and (**a**) LC3B (blue) (*n* = 3) or (**b**) Beclin1 (*n* = 4). (**c**) c-Met-GFP (green) cells, plated on poly-L-lysine, incubated with tetracycline (Tet) for 16 h and stained for β1-integrin (red) and LC3B (blue) (*n* = 3). (**d,e**) MDA-MB-468 cells at 120 min HGF stimulation in suspension, stained for c-Met (red), β1-integrin (green) and (**d**) LC3B (blue) and DAPI (magenta) (*n* = 4) or (**e**) Beclin1 (blue) and DAPI (magenta) (*n* = 3). (**f,g**) Western blots for (**f**) β1-integrin, c-Met, ATG5, tubulin and phospho-ERK1/2 or (**g**) c-Met, ATG13, tubulin and phospho-ERK1/2 in A549 cells stimulated without ( − ) or with ( + ) HGF for 15 or 120 min in suspension. Numbers represent mean phospho-ERK1/2/tubulin ratios ± s.e.m. at 120 min with HGF normalized to no HGF, in cells transfected with: (**f**) control or ATG5 (SMARTpool) siRNA (*n* = 3); (**g**) control or ATG13 siRNA (from two to four individual oligos used per experiment, pooled data from one or multiple individual oligos per experiment, *n* = 5). Data were obtained by densitometric analysis. (**h,i**) Western blots for (**h**) β1-integrin, c-Met, ATG5, tubulin and phospho-ERK1/2 or (**i**) c-Met, ATG13, Beclin1, tubulin and phospho-ERK1/2 in c-Met-GFP cells stimulated without ( − ) or with ( + ) tetracycline (Tet) for 16 h. Numbers represent mean phospho-ERK1/2/tubulin ratios ± s.e.m. normalized to control knocked down cells + Tet in (**h**) ATG5 (SMARTpool) knocked down c-Met-GFP cells (*n* = 3) or (**i**) ATG13 or Beclin1 knocked down (two individual oligos of each) c-Met-GFP cells + Tet (*n* = 3). Data were obtained by densitometric analysis. (**j**) Western blot for c-Met, ATG13, tubulin and phospho-ERK1/2 in ATG13 knockout MEFs re-expressing (WT) or not ( − / − ) WT ATG13 stimulated without ( − ) or with ( + ) HGF for 120 min in suspension. Numbers represent mean phospho-ERK1/2/tubulin ratios ± s.e.m. at 120 min with HGF normalized to no HGF (*n* = 3). Data were obtained by densitometric analysis. (**k**) Confocal sections of c-Met-GFP (green) cells with tetracycline (Tet) for 0 and 16 h. Cells were stained for phospho-ERK1/2 (red), LC3B (blue) and DAPI (purple). Scale bar, 10 μm. Numbers are percentages colocalization ± s.e.m. between c-Met-LC3B and P-ERK1/2 (*n* = 3). (**l,m**) Percentage of the colocalization of P-ERK1/2 with c-Met-LC3B in (**l**) c-Met-GFP cells upon Tet transfected with β1-integrin (β1) siRNA and (**m**) β1A-YYFF cells upon HGF for 120 min, compared with the colocalization in their respective control, set as 100%. Controls are siRNA control + Tet (**l**) (Cont) and β1A cells + HGF 120 min (**m**). Data are mean ± s.e.m. (*n* = 3). *t*-Test, *$P < 0.05$; **$P < 0.01$; ***$P < 0.001$.

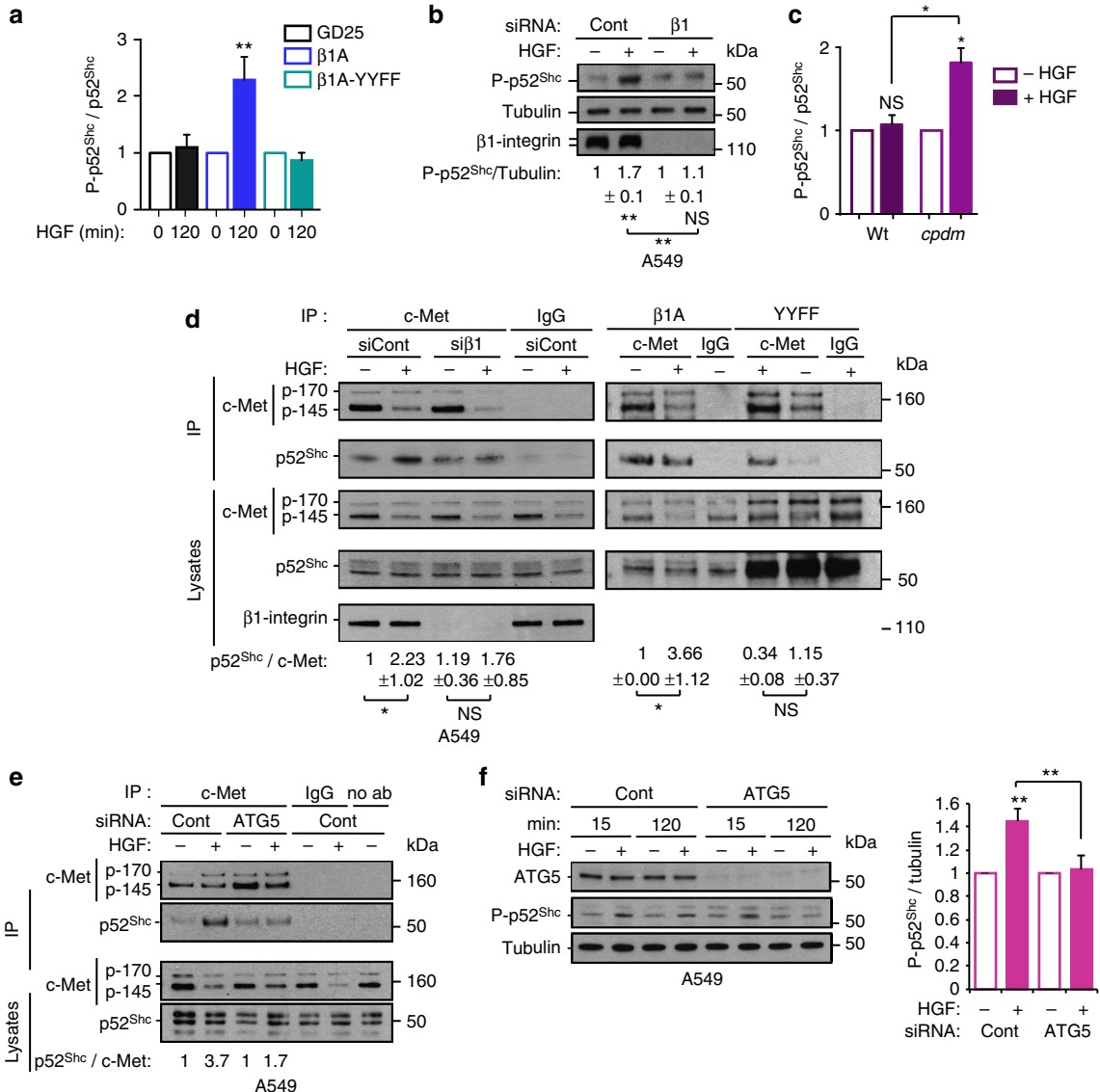

**Figure 6 | β1-integrin plays the role of an adaptor to sustain c-Met signalling on ARE.** (**a**) Graph represents mean phospho-p52$^{Shc}$/p52$^{Shc}$ ratios ± s.e.m. upon 120 min HGF stimulation normalized to 0 min, in GD25, β1A and β1A-YYFF cells. Data were obtained by densitometric analysis of western blots (shown in Supplementary Fig. S7a) ($n = 3$ GD25 cells, $n = 4$ β1A and β1A-YYFF cells). (**b**) Western blots for phospho-p52$^{Shc}$, tubulin and β1-integrin in A549 cells, transfected with control (Cont) or β1-integrin (β1) siRNA and stimulated without ($-$) or with ($+$) HGF for 120 min in suspension. Numbers are mean phospho-p52$^{Shc}$/tubulin ratios ± s.e.m. upon HGF stimulation normalized to no HGF, obtained by densitometric analysis ($n = 3$). (**c**) WT and cpdm (Sharpin null) MEFs were incubated without ($-$) or with ($+$) HGF for 120 min in suspension. Graph represents the mean phospho-p52$^{Shc}$/p52$^{Shc}$ ratios ± s.e.m. with HGF normalized to no HGF obtained by densitometric analysis of western blots (shown in Supplementary Fig. 6c) ($n = 3$). (**d**) Western blots for c-Met and p52$^{Shc}$ following immunoprecipitations with c-Met (CVD13 for left panel, B2 for right panel) or IgG control. Total c-Met, p52$^{Shc}$ and/or β1-integrin levels in the cell lysates are shown. All cells were treated without ($-$) or with ($+$) HGF for 120 min. Left panel: A549 cells transfected with control or β1-integrin siRNA and maintained in suspension; Right panel: β1A and β1A-YYFF cells. Numbers represent the ratios of p52Shc co-immunoprecipitated with c-Met (normalized on the IgG values) ± s.e.m. ($n = 3$). Left panel: values are fold change versus siCont-HGF. Right panel: values are fold change versus β1A cells- HGF. (**e**) Western blots for c-Met and p52$^{Shc}$ following immunoprecipitations with c-Met (CVD13), IgG control or no antibody (no ab). A549 cells were transfected with control or ATG5 siRNA without ($-$) or with ($+$) HGF for 120 min in suspension. Total c-Met and p52$^{Shc}$ in the cell lysates are shown. (**f**) Western blots for ATG5, phospho-p52$^{Shc}$ and tubulin in A549 cells in suspension without ($-$) or with ($+$) HGF for 15 or 120 min. Graph represents mean phospho-p52$^{Shc}$/tubulin ratios ± s.e.m. at 120 min with HGF, normalized to no HGF at 120 min, in control (Cont) and ATG5 siRNA-transfected cells, obtained by densitometric analysis ($n = 4$). t-Test, *$P < 0.05$; **$P < 0.01$; NS: not significant.

activated constantly upon HGF stimulation (Fig. 6a,b, Supplementary Fig. 7a,b). The absence (GD25 cells or β1-integrin knocked down A549 and c-Met-GFP cells) or mutation (β1A-YYFF cells) of β1-integrin significantly impaired c-Met-dependent sustained p52$^{Shc}$ phosphorylation (at 120 min of HGF stimulation or after 16 h of tetracycline treatment), compared with control (β1A or control knocked down A549 and

c-Met-GFP cells) (Fig. 6a,b, Supplementary Fig. 7a,b). HGF significantly stimulated p52$^{Shc}$ phosphorylation at 120 min in cpdm cells while the signal was not sustained in WT cells (Fig. 6c, Supplementary Fig. 7c). The depletion of p52$^{Shc}$ using siRNA, reduced c-Met-dependent sustained ERK1/2 phosphorylation (shown at 120 min of HGF stimulation) in A549 and MDA-MB-468 (Supplementary Fig. 7d).

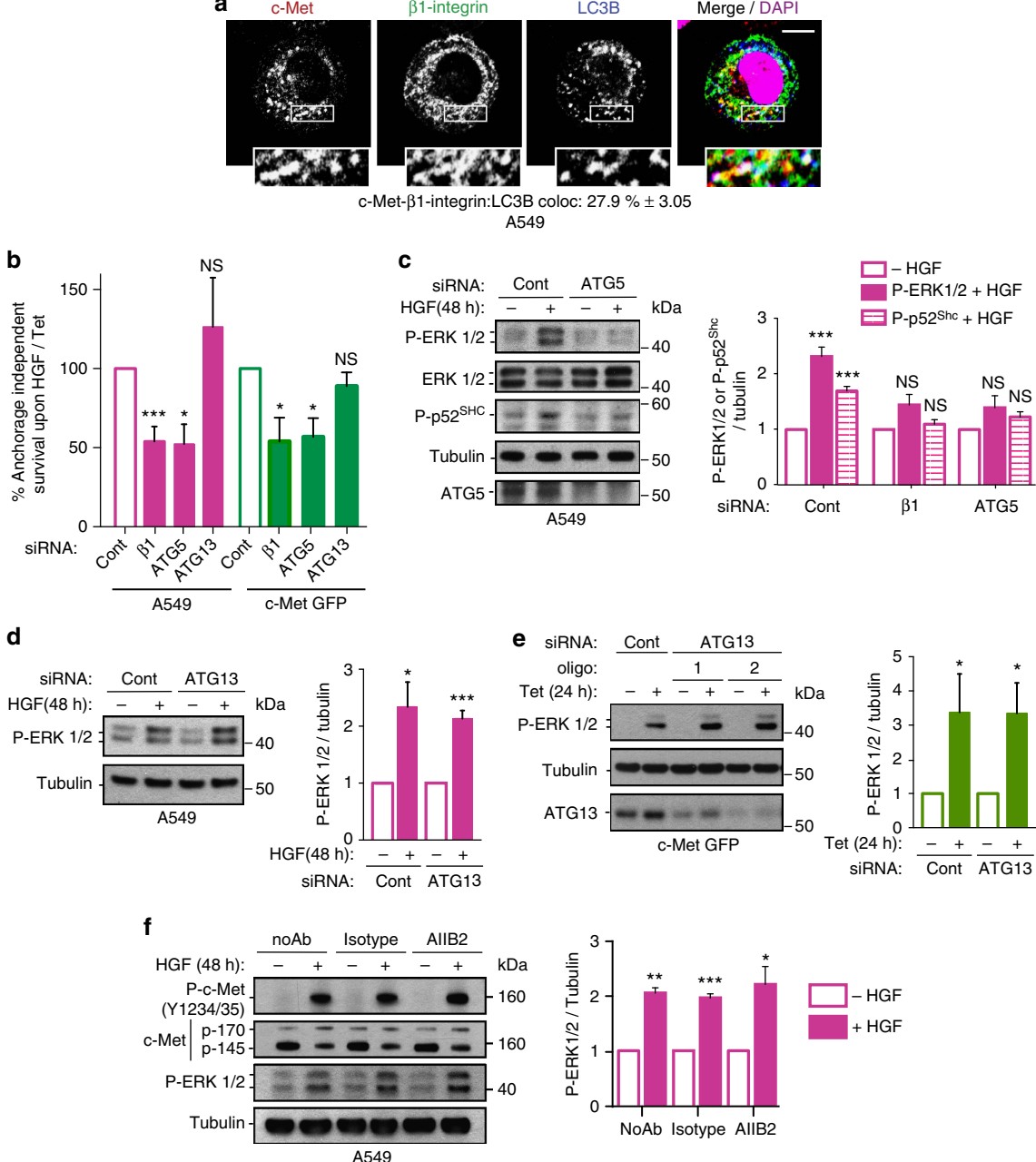

**Figure 7 | c-Met–β1-integrin cooperation on ARE mediates c-Met-dependent anchorage-independent cell survival.** (**a**) Confocal section of A549 cells after HGF stimulation in suspension for 48 h. Cytospun and fixed cells were stained for c-Met (red), β1-integrin (green), LC3B (blue) and DAPI (magenta). Scale bar, 10 μm. Numbers are mean percentage colocalization ± s.e.m. between c-Met–β1-integrin and LC3B (n = 3). (**b**) Mean percentage of HGF- or tetracycline-dependent anchorage-independent survival (or protection against anoikis) ± s.e.m. in cells transfected with β1-integrin (β1)(Qiagen), ATG5 (SMARTpool) or ATG13 (from one to four individual oligos used per experiment, pooled data from one or multiple individual oligos per experiment) siRNA, normalized to control (Cont) siRNA. A549 cells were stimulated with HGF for 48 h in suspension. c-Met-GFP cells were stimulated with tetracycline (Tet) for 24 h in suspension. The HGF- or Tet-dependent anchorage-independent survival was obtained by normalizing the data with HGF/Tet to no HGF/Tet. Cells were stained with propidium iodide and the cell viability was analysed by flow cytometry (A549: β1-integrin siRNA n = 6, ATG5 or ATG13 siRNA n = 3. c-Met-GFP: β1-integrin or ATG5 siRNA n = 3, ATG13 siRNA n = 4). (**c**) Western blots for phospho-ERK1/2, ERK1/2, phospho-p52Shc, tubulin and ATG5 in A549 cells, transfected with control or ATG5 siRNA and stimulated without (−) or with (+) HGF for 48 h in suspension. Graph represents mean ratios ± s.e.m. of phospho-ERK1/2/tubulin (siRNA β1-integrin: n = 5, siRNA ATG5: n = 3) and of phospho-p52Shc/tubulin (siRNA β1-integrin: n = 3; siRNA ATG5: n = 4) with HGF normalized to the mean ratios with no HGF, obtained by densitometric analysis. (**d,e**) Western blots for (**d**) phospho-ERK1/2 and tubulin or (**e**) phospho-ERK1/2, tubulin and ATG13 in (**d**) A549 cells stimulated without (−) or with (+) HGF for 48 h in suspension or (**e**) c-Met-GFP cells stimulated without (−) or with (+) Tet for 24 h in suspension, transfected with control or (**d**) one (oligo 3) or (**e**) two (oligo 1 and 2) individual ATG13 siRNA oligos. Graph represents mean fold increases of phospho-ERK1/2/tubulin (from one to four individual oligos used per experiment, pooled data from one or multiple individual oligos per experiment) upon HGF/Tet versus no HGF/Tet ± s.e.m. (n = 4), obtained by densitometric analysis. (**f**) Western blots for phospho-c-Met (Y1234-35), c-Met, phospho-ERK1/2 and tubulin in A549 cells pre-treated with the AIIB2 β1-integrin blocking antibody (2 μg ml⁻¹), an isotype control or no antibody, and stimulated without (−) or with (+) HGF for 48 h in suspension. Graph represents mean phospho-ERK1/2/tubulin ratios ± s.e.m. with HGF normalized to the ratios with no HGF obtained by densitometric analysis (n = 3). t-Test, *P < 0.05; **P < 0.01; ***P < 0.001.

The HGF-stimulated increase of p52[Shc] co-immunoprecipitation with c-Met was reduced significantly upon β1-integrin knockdown (in detached A549 cells) or mutation (β1A-YYFF cells) compared with controls (control knockdown or β1A cells) (Fig. 6d). A ternary complex p52[Shc]–c-Met–β1-integrin was detected post β1-integrin (Supplementary Fig. 7e) or c-Met (Supplementary Fig. 7f) immnunoprecipitation. ATG5 siRNA knockdown impaired the HGF-dependent co-immunoprecipitation of p52[Shc] with c-Met (Fig. 6e) and with β1-integrin (Supplementary Fig. 7e) in detached A549 cells. Accordingly, sustained (but not transient) c-Met-dependent p52[Shc] phosphorylation (in A549 cells at 120 min of HGF stimulation and in 16 h tetracycline stimulated c-Met-GFP cells, each in suspension) was reduced significantly upon ATG5 siRNA knockdown (Fig. 6f, Supplementary Fig. 7g). Moreover, while ATG5 siRNA knockdown in *cpdm* cells had no influence on p52[Shc] phosphorylation activation levels at 15 min of HGF, it reduced p52[Shc] phosphorylation at 120 min (Supplementary Fig. 7h). Furthermore, phosphorylated p52[Shc] colocalization with the pool of c-Met within LC3B vesicles upon HGF/tetracycline treatment was reduced upon β1-integrin knockdown (Supplementary Fig. 7i,j).

Altogether, these results suggest that on ARE, active conformation β1-integrin, through its NXXY domain, plays the role of a scaffold between c-Met and p52[Shc], leading to sustained ERK1/2 activation.

**c-Met–β1-integrin signal on ARE for cell survival in anoikis.** As c-Met–β1-integrin cooperation occurs in detached cells, we investigated whether it plays a role in c-Met-dependent anchorage-independent survival (or anoikis resistance). Cell death under anchorage-independent cell culture conditions was investigated for the various cell lines (see Methods section). Thus 40–70% of cells maintained under non-adherent conditions for 24 h (NIH3T3 c-Met WT/M1268T, c-Met-GFP) or 48 h (MDA-MB-468 and A549) were propidium iodide positive (PI). Tetracycline, HGF treatment or M1268T mutation significantly decreased the percentage of PI-positive cells (Supplementary Fig. 8a–d). Under such conditions, a pool of internalized c-Met colocalized with β1-integrin on ARE, as indicated by triple colocalization with LC3B (Fig. 7a, Supplementary Fig. 8e and Supplementary Data 1) while c-Met and β1-integrin could be immunoprecipitated (Supplementary Fig. 8f).

Strikingly β1-integrin or ATG5/Beclin1 knockdown, but not ATG13 knockdown, significantly reduced c-Met-dependent survival (Fig. 7b, Supplementary Fig. 8g,h,j), with no effect on basal cell death levels (Supplementary Fig. 8i); and reduced c-Met-dependent ERK1/2 and p52[Shc] activation (Fig. 7c–e, Supplementary Fig. 8k–r) with unchanged c-Met and β1-integrin expression or c-Met phosphorylation levels (Supplementary Fig. 8k,l).

Independence of β1-integrin ligand on c-Met-dependent ERK1/2 phosphorylation was confirmed using the β1-integrin blocking antibody AIIB2 (Fig. 7f). Importantly, MEK inhibition impaired the c-Met-dependent increase in cell survival (Supplementary Fig. 8s). Clathrin siRNA knockdown reduced HGF-dependent anchorage-independent survival in A549 cells, confirming the requirement for endocytosis (Supplementary Fig. 8s). Inhibition of recycling using Primaquine or RCP siRNA[43] actually increased this survival advantage (Supplementary Fig. 8s). Thus, c-Met and β1-integrin recycling per se does not determine cooperation leading to cell survival; rather it is their intracellular localization on ARE which is important.

**c-Met–β1-integrin inside-in signalling mediates tumorigenesis.** We analysed the role of β1-integrin in c-Met-dependent anchorage-independent growth in soft agar. The absence of β1-integrin (siRNA knockdown in c-Met M1268T expressing cells or GD25 cells) significantly decreased c-Met dependent anchorage-independent growth in soft agar, compared with controls (control siRNA and β1A cells) (Fig. 8a and Supplementary Fig. 9a). Pharmacological MEK inhibition by U0126 also inhibited the HGF effect on β1A colony sizes (Supplementary Fig. 9b,c).

We investigated whether β1-integrin–c-Met cooperation occurs inside the cells to stimulate anchorage-independent growth, *in vivo* tumorigenesis and invasion, through assessing the role of β1-integrin NXXY motifs. As for GD25 cells, HGF did not increase the size of colonies in soft agar formed by β1A-YYFF cells (Fig. 8a). GD25, β1A and β1A-YYFF cells were then grown in soft agar +/– HGF-secreting MRC5 fibroblasts[24], +/– the c-Met inhibitor PHA-665752. MRC5 cells had no effect on colony area formed by GD25 and β1A-YYFF cells but increased the colony area formed by β1A cells. Moreover, PHA-665752 reduced β1A cell colony area (Supplementary Fig. 9d). GD25 or β1A or β1A-YYFF cells were grafted subcutaneously into nude mice together with MRC5 cells. PHA-665752 applied daily (topically) to the growing tumours[6], reduced β1A but not GD25 and β1A-YYFF tumours (Fig. 8b). When injected into zebrafish embryos together with HGF, +/– PHA-665752 to control for c-Met activity, β1A cells were significantly more invasive than β1A-YYFF cells; PHA-665752 inhibited the invasion of β1A but not of β1A-YYFF cells (Fig. 8c). Finally, ATG5 siRNA knockdown significantly reduced invasion of the A549 cells incubated with HGF in zebrafish embryos, suggesting further that β1-integrin–c-Met cooperation occurs on 'ARE' *in vivo* (Fig. 8d).

In summary, β1-integrin-dependent c-Met signalling promotes anchorage-independent survival and growth, tumour growth and metastasis, and occurs inside the cells. This novel β1-integrin signalling supports c-Met-dependent survival in anchorage independence conditions via a ligand- and adhesion-independent scaffolding function, mediating p52[Shc] and ERK1/2 pathway activation. We propose that β1-integrin triggers an 'inside-in signalling' on the 'ARE', leading to c-Met-sustained signalling, promoting cell survival in anchorage-independent growth conditions, leading to enhanced metastasis.

## Discussion

Our study reveals a novel non-adhesive function of β1-integrin in cooperation with the RTK c-Met that we call 'an inside-in signalling'. This pathway leads to c-Met-dependent cell anchorage-independent survival/growth, *in vivo* tumorigenesis, invasion and metastasis. It occurs on LC3B-positive endomembranes, belonging to a non-canonical autophagy pathway. These AREs represent novel RTK signalling platforms.

We show for the first time that β1-integrin is a major regulator of the c-Met pathway at two levels: (i) determining optimal internalization of activated c-Met (as shown for PDGFR[17]) with co-internalization of the two molecules; (ii) post-internalization, on ARE, β1-integrin promotes c-Met-dependent sustained p52[Shc] and ERK1/2 signalling, likely through acting as a scaffold linking c-Met to p52[Shc].

This novel β1-integrin-dependent c-Met signalling occurs in detached cells and is independent of integrin adhesive properties. Interestingly, β4-integrin was reported to mediate RTK signalling by acting as a scaffold, independently of ligand binding, though no known link with trafficking, of either the RTK or β4-integrin[44], was reported.

An active β1-integrin conformer, as triggered by the absence of the endogenous β1-integrin inhibitor SHARPIN or incubation of cells in MnCl₂, appears to be required, as c-Met-sustained ERK1/2 signalling is enhanced in these conditions. Recently, trafficked

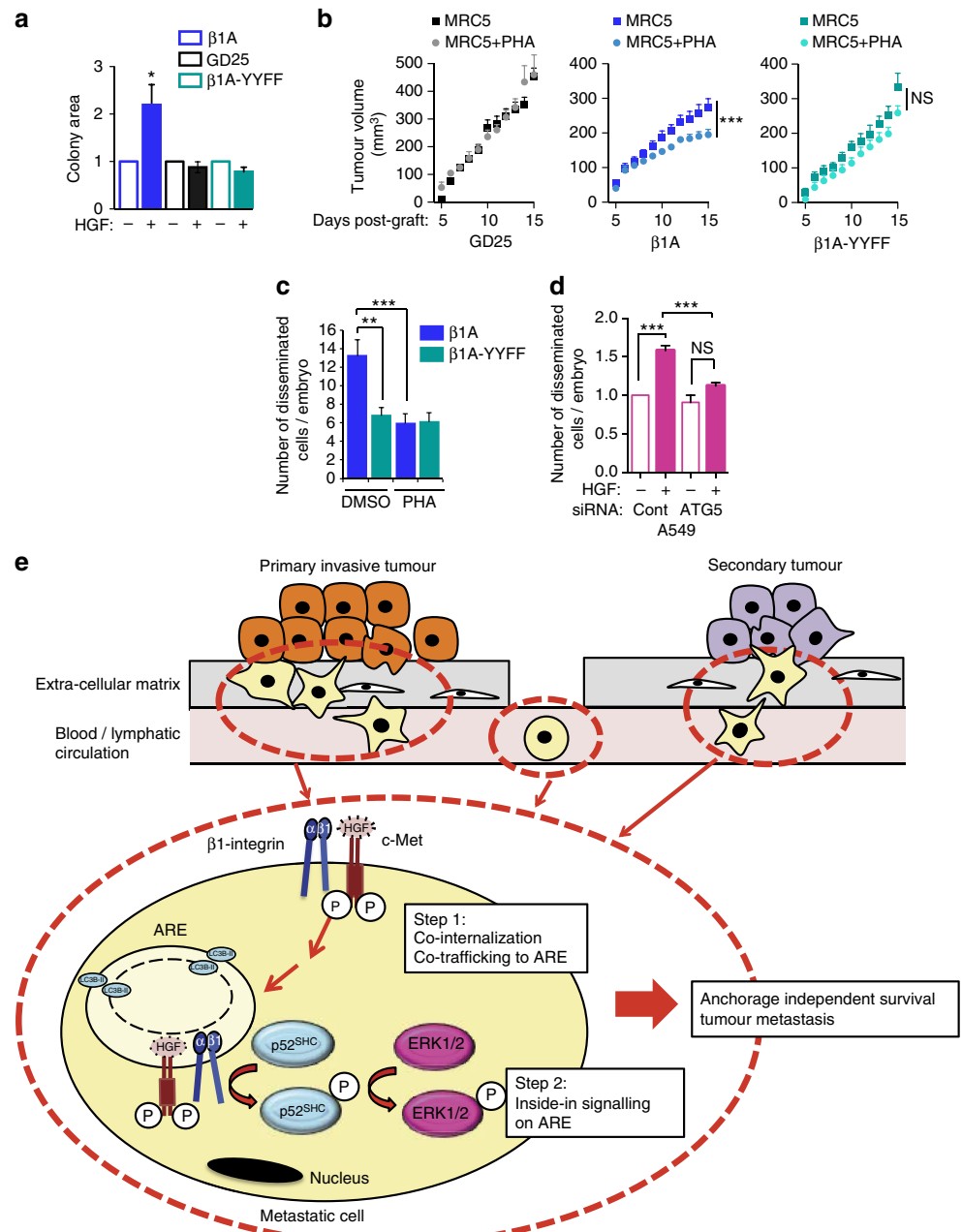

**Figure 8 | c-Met–β1-integrin intracellular cooperation mediates c-Met-dependent anchorage-independent growth, *in vivo* tumorigenesis and invasion.** (**a**) Mean area of β1A, GD25 and β1A-YYFF colonies treated with (+) HGF in soft agar normalized to the mean area without HGF (−) ± s.e.m. (*n* = 3, each experiment performed in duplicate). (**b**) Tumour growth curves, over time, of GD25, β1A and β1A-YYFF cells mixed with MRC5 fibroblasts and treated daily with DMSO or PHA-665752 (PHA, 100 nM) by topical application onto the surface of the skin where cells had been injected from day 1 after injection. Graphs represent the mean tumour volumes (mm³) ± s.e.m. of *n* = 5 mice per group measured daily. (**c**) Mean number of disseminated β1A and β1A-YYFF cells per zebrafish embryo 24 h after injection in the yolk sac ± s.e.m. Cells were incubated with HGF and treated with DMSO or PHA-665752 (PHA, 100 nM) (*n* = 3, average of 26 embryos per condition per experiment). (**d**) Mean number of disseminated A549 cells per embryo 24 h after injection in the yolk sac ± s.e.m. Cells were transfected with control or ATG5 siRNA and incubated without (−) or with (+) HGF (*n* = 3, average of 39 embryos per condition per experiment). (**e**) Model of β1-integrin–c-Met 'inside-in' signalling, promoting survival of cancer cells during metastasis and in the establishment of tumours: In unstimulated cells, c-Met and β1-integrin form a complex at the plasma membrane. c-Met activation in either a ligand-dependent (A549/MDA-MB-468/MEF cells) or -independent (c-Met-GFP/M1268T c-Met-expressing NIH3T3 cells) manner results in internalization and activation of β1-integrin, which is the major form to internalize with c-Met. β1-integrin and its active conformation are in turn required for an optimal endocytosis of c-Met. Thus, they both need each other for their optimal endocytosis. They co-internalize in a clathrin-dependent manner. Internalized c-Met-β1-integrin complex progressively accumulates on 'autophagy-related endomembranes' (ARE), which are LC3B-positive endomembranes. β1-integrin promotes sustained c-Met signalling from ARE likely through acting as an adaptor that links c-Met to p52Shc, which in turn activates the downstream signalling pathway ERK1/2. Altogether, c-Met-β1-integrin cooperation, that we named 'inside-in signalling' is required for anchorage-independent survival that may help cancer cells to survive as they invade from the primary tumour, travel from the primary tumour to the secondary site during metastasis, leading to tumour growth and metastasis. *t*-Test for **a** and **c**; ANOVA test for **b**; Mann–Whitney *U*-test for **d**; *P < 0.05; **P < 0.01; ***P < 0.001; NS: not significant.

active β1-integrin was shown to reside significantly longer on the endosomes compared with the fast recycling inactive β1-integrins[28], consistent with our notion of sustained signalling of c-Met from the 'ARE'.

The classical function of autophagosomes, through fusion with lysosomes, is to promote degradation of intracellular materials and organelles, maintaining cellular homoeostasis[40,45]. Recently however, phosphorylated Src was shown to localize on autophagosomes in FAK−/− cells, leading to its degradation and cell survival[46]. Upon EGF stimulation, ERK1/2, and its upstream kinase MEK, localize to pre-autophagosomes and autophagy proteins promote ERK1/2 phosphorylation[41].

We propose that 'ARE' represent novel platforms for efficient spatial coordination of signalling cascades. Our results suggest that although c-Met–β1-integrin form a complex independently of their localization on 'ARE', the complex needs to localize to 'ARE' to allow the β1-integrin scaffolding function. Furthermore our data indicate that c-Met activation and c-Met and β1-integrin stability are not influenced by autophagy while c-Met activity, or β1-integrin expression and NXXY domain, have no role on autophagy. β1-integrin-LC3B co-localization has been reported previously[47], however, this is the first report of c-Met localization on LC3B and Beclin1-positive endomembranes, together with β1-integrin. Thus, c-Met and β1-integrin co-internalize, and progressively accumulate on LC3B and Beclin1-positive vesicles, 'ARE' (optimal enrichment at 120 min of HGF stimulation). Although 'ARE' require further characterization, our results suggest that these endomembranes belong to a recently described non-canonical autophagy pathway, which was so far shown associated with macroendocytic engulfment processes[34–37]. This highlights a novel and unexpected intracellular localization for c-Met signalling.

Metastatic epithelial cancer cells detach from the ECM and survive, due to their anchorage-independent properties, for sufficient time to facilitate distal colonization[48]. Our results suggest that, in detached cells, c-Met uncouples survival from adhesion and uses β1-integrin as an adaptor to amplify c-Met signalling to ERK1/2, on 'ARE', leading to increased survival (see model in Fig. 8e). Thus, this signalling may occur during specific time-windows of the metastatic process in addition to the classical adhesive property of β1-integrin, which also likely cooperates with c-Met during the metastatic process such as during cell invasion.

Integrins are considered to be important cancer therapy targets and several inhibitors, which alter the adhesive property of integrins[49–53], are being tested in the clinic. The results presented here suggest that β1-integrin also contributes to cancer metastasis using signalling properties independent from its adhesive function. This suggests that alternative therapies to fully alter integrin functions are needed. Targeting integrin signalling in addition to adhesion may have relevance for cancer therapy. Present models of integrin 'inside-out' and 'outside-in' signalling may need to be refined to encompass the potential contribution of 'inside-in signalling'.

## Methods

**Cell lines and cell culture.** The β1-integrin-deficient GD25 cell line (GD25), GD25 cells expressing wild-type β1A (β1A) or mutant β1A (β1A-YYFF) (gift from S. Johansson)[23] were cultured in Dulbecco's modified Eagle's medium (DMEM) containing 10% fetal bovine serum (FBS, Sigma) and 2 mM L-glutamine, with the addition of puromycin (5 μg ml$^{-1}$) for the β1A and β1A-YYFF cells. For stimulation experiments GD25 cell lines were plated in 6-well plates at $2.5 \times 10^5$ cells per well for 48 h. Twenty-four hours before stimulation, the cells were starved in serum-free medium and stimulated in serum-free medium with 50 ng ml$^{-1}$ of HGF for the times indicated.

T-REx-293 cell line (Invitrogen Life Technologies) was maintained in DMEM containing 10% FBS and 5 μg ml$^{-1}$ blasticidin (Invitrogen Life Technologies). T-REx-293 cell line was transfected with Lipofectamine 2000 (Invitrogen)

according to the manufacturer's instructions. T-REx-293 cell line stably transfected with c-Met-GFP ('c-Met-GFP cells') was established by selection with Zeocin (400 μg ml$^{-1}$; Invitrogen Life Technologies) following the manufacturer's instructions. Such cells were maintained in culture as the T-REx-293 cell line with the addition of 400 μg ml$^{-1}$ of Zeocin. Expression of c-Met-GFP was induced by treating the stable cell line with 0.1 μg ml$^{-1}$ of tetracycline for indicated times (16 h for most experiments).

NIH3T3 cells expressing c-Met WT or M1268T murine cDNA were a gift from Prof. G. Vande Woude and were cultured in DMEM containing 10% donor calf serum (Gibco Life Technologies)[6,54].

A549 cells (ATCC) were cultured in DMEM containing 10% FBS and 2 mM L-glutamine.

MDA-MB-468 cells (ATCC) were cultured in phenol-free DMEM containing 10% FBS and 2 mM L-glutamine.

MRC5 cells (ATCC) were maintained in Minimum essential medium containing 10% FBS. Conditioned media was taken after three days of culture when cells were 70% confluent.

ATG13 knockout mouse embryonic fibroblasts (MEFs) and the reconstituted Flag-S-tagged WT ATG13 MEFs were a gift from Dr. Noor Gamooh and were previously published[38]. The MEFs were cultured in DMEM containing 10% FBS and 2 mM L-glutamine.

The Flag-S-tagged WT ATG13 MEFs were cultured with puromycin (1 μg ml$^{-1}$).

All cells were maintained at 37 °C in a humidified 8% CO$_2$ atmosphere.

**Constructs.** The human hepatocyte growth factor receptor (c-Met) open reading frame (ORF) was first introduced into pEGFP-N1 (BD Clontech). c-Met ORF was amplified by PCR, using the flanking primers 5′-ccgctcgagatgaaggcccccgctgtgc-3′ (Xho I site) and 5′-ccccaagcttcaatgatgtctcccagaaggaggc-3′ (Hind III site) (the underlined sequence represents the mutated stop codon). Plasmid containing c-Met-EGFP construct was then digested with Eco RI and NotI restriction enzymes and was introduced into pcDNA 4/TO (Invitrogen Life Technologies) containing full-length WT c-Met digested with both restriction enzymes. Construct was checked by full sequencing. GFP-Rab21 construct was described[31] and α5-integrin-GFP construct was provided by Horwitz[55].

**Reagents.** Purified human recombinant HGF was obtained from R&D Systems and used at 50 ng ml$^{-1}$ (all experiments using the GD25 cell model and 48 h stimulations in all cell lines) or 100 ng ml$^{-1}$ (all other experiments).

HGF-AlexaFluor-555 was generated using the Alexa Fluor 555 Microscale Protein Labelling kit (Thermofisher) according to the manufacturer's instructions.

The following antibodies were used:

—Mouse monoclonals anti-: GFP (CR-UK), LC3B (clone 5F10, Novacastra), tubulin (Sigma-Aldrich), mouse c-Met extracellular domain (B2, sc-8057, Santa Cruz).

—Rabbit polyclonals anti-: human c-Met intracellular domain (sc-10, Santa Cruz Biotechnologies and CVD13, Invitrogen), phospho-c-Met (Tyrosine 1349 or Tyrosine 1234/1235, Cell Signalling), phospho-ERK1/2 (Cell Signalling and R&D Systems (MAB1018)), pan-ERK1/2 (Upstate), phospho-SHC Y239/240 (CS2434, Cell Signalling), pan-SHC (CS2432, Cell Signalling), ATG5 (TMD-PH-AT5, Cosmo Bio Co Ltd.), LC3B (CS2775, Cell signalling), early endosome antigen 1 (EEA1) (Santa Cruz Biotechnology), Beclin1 (CS3738, Cell Signalling), ATG13 (SAB4200100, Sigma-Aldrich).

—Goat polyclonals anti-: human c-Met extracellular domain (AF276, R&D Systems), mouse c-Met extracellular domain (AF527, R&D Systems), early endosome antigen 1 (EEA1) (Santa Cruz Biotechnology).

—Rat monoclonal anti-mouse β4-integrin (553745, BD Biosciences).

The following β1-integrin antibodies were used:

—Mouse monoclonals anti-: human β1-integrin, clone DF7 (Enzo Life Sciences), human β1-integrin (MAB2252, Millipore), human CD29 clone K20 (Beckman Coulter) labelled with AlexaFluor-488.

—Rat monoclonals anti-: mouse β1-integrin, clone MB1.2 (MAB1997, Millipore), β1-integrin in active conformation, clone 9EG7 (BD Biosciences).

—Rat polyclonal anti-: β1-integrin AIIB2 (Developmental Studies Hybridoma Bank).

—Rabbit polyclonal anti-: human β1-integrin AB1952 (Millipore).

The following blocking β1-integrin antibodies were used:

—Rat polyclonal anti- β1-integrin AIIB2 (Developmental Studies Hybridoma Bank) at 2 μg ml$^{-1}$ and LEAF purified anti-mouse CD29 Armenian hamster IgG clone HMβ1-1 at the concentration indicated.

The secondary antibodies used for Western blot were peroxidase-labelled sheep anti-mouse, donkey anti-rabbit IgG or goat anti-rat IgG (VWR international) used at 1:1,000.

The secondary antibodies used for immunofluorescence experiments were Alexa 488-conjugated donkey anti-rabbit/mouse/goat/rat IgG, (Molecular Probes, Life Technologies), Cy3- or Cy5-conjugated affinity-purified donkey anti-mouse/rabbit/goat/rat IgG (Jackson ImmunoResearch) used at 1:500.

The secondary antibodies used for FACS were PE-, APC-conjugated (Becton Dickinson) or AlexaFluor-488 conjugated (Molecular Probes, Life Technologies) used at 1:250.

Fibronectin (from bovine plasma) (1:100), Laminin (1:100) and poly-L-lysine (0.01%) were obtained from Sigma-Aldrich and used to coat the wells.

Prolong gold mounting media containing DAPI was obtained from Life Technologies.

Poly-L-lysine, tetracycline, Dynasore, PHA565752 (PHA) and $MnCl_2$ were obtained from Sigma-Aldrich. LY294002, SU1498 and SU11274 were obtained from Calbiochem.

**Transfections of cDNA and RNAi.** Transfections of cDNA constructs were carried out using Lipofectamine 2000 (Invitrogen Life Technologies) as described previously[3] or by electroporation using Amaxa Nucleofactor Technology following the manufacturer's instructions (Lonza).

Transfections of siRNA were carried out using oligofectamine (Invitrogen Life Technologies) as described previously[3], using HiPerFect reagent (Qiagen) or by electroporation using Amaxa Nucleofactor technology following the manufacturer's instructions (Lonza). Cells were harvested or subjected to experimental procedures 72 h after transfection unless otherwise stated. NIH3T3 cells were an exception with experimental procedures conducted 48 h after transfection unless otherwise stated.

See Supplementary Table 1 for details on the siRNA target sequences used in this study.

**Cell stimulation in suspension and anoikis assay.** Cells were cultured for 3 days on plastic (after transfection with siRNA for some experiments), detached using trypsin, harvested with 0.2% Soya bean trypsin inhibitor (Sigma-Aldrich) in serum-free media, then washed in serum-free media and centrifuged.

For cell stimulation in suspension, $3 \times 10^5$ cells were transferred to 2 ml eppendorf tubes in 500 μl of serum-free media, maintained at 37 °C for 3 h and then stimulated with 100 ng per ml HGF for 120 min. Cells were put on ice and harvested for western blot or cytospun and fixed for immunofluorescence.

For the anoikis assay $1 \times 10^6$ cells were transferred to 50 ml falcon tubes for 24 h (c-Met-GFP and NIH3T3 cells) or 48 h (MDA-MB-468 and A549 cells) in 10 ml of serum-free media (MDA-MB-468, NIH3T3 and c-Met-GFP) or full serum (A549) media +/− tetracycline (c-Met-GFP) or ±50 ng ml$^{-1}$ HGF (A549 and MDA-MB-468) at 37 °C. Cells were put on ice and either harvested for western blot, cytospun and fixed for immunofluorescence, or stained with propidium iodide (1/100, Life Technologies) for 15 min. The percentage of dead cells was determined by measuring those cells that could incorporate propidium iodide, using a FACS Calibur.

**Western blot analysis.** Cell were harvested in radioimmunoprecipitation buffer (RIPA) or directly in Laemmli sample buffer (Invitrogen) and boiled for 10 min. Samples were loaded on 4–12% gradient polyacrylamide gels (Invitrogen). Separated proteins were transferred to a 0.45-mm nitrocellulose transfer membrane (Whatman). Protein loading was checked by staining with Ponceau Red. Membranes were then blotted with appropriate first antibodies at a dilution of 1:1,000. Specific binding of antibodies was detected with appropriate peroxidase-conjugated secondary antibodies and visualized by enhanced chemiluminescence detection (GE Healthcare)[5]. Densitometric analyses of immunoblots were performed using ImageJ 1.47v (National Institute of Health). Full blots are included in the Supplementary Information (Supplementary Figs 10 and 11).

**Co-immunoprecipitations.** Following treatment with HGF for the indicated time, and in cells in suspension when indicated in the figure legend, cells were placed on ice, washed with cold PBS and lysed in a buffer containing 0.5% Triton X-100, 20 mM Tris pH7.5, 150 mM NaCl, and protease and phosphatase inhibitors in PBS. Cell lysates were collected, rotated for 30 min at 20 r.p.m. on a wheel at 4 °C, centrifuged for 3 min at 2,000 r.p.m. at 4 °C, and the supernatants were collected. Fifty microliter of the supernatant was reserved for total input. The remaining lysates were pre-cleared by adding 25 μl of washed A/G agarose beads for 1 h at 4 °C on a rotating wheel. The pre-cleared lysates were centrifuged and the supernatant transferred into new tubes. Two microgram of antibody (c-Met: B2 anti-mouse; β1-integrin: MAB1997, Millipore), 25 μl of washed A/G beads were added and the samples were rotated for 2 h at 4 °C. The lysates were centrifuged, the pellets were collected and washed three times with lysis buffer and three times with wash buffer: 20 mM Tris pH7.5, 150 mM NaCl, and protease and phosphatase inhibitors. The samples were analysed by Western blotting. Quantifications were obtained by densitometric analysis of the Western blots. Values were first thresholded on IgG values and then normalized on the levels of the immunoprecipitated protein (c-Met or β1-integrin). The levels of co-immunoprecipitation at 0 min HGF was set as 1 and the fold change in levels upon HGF was shown.

**Immunofluorescence and confocal microscopy.** Cells ($5 \times 10^4$) were plated onto coverslips coated with 0.01% poly-L-lysine (Sigma). Immunofluorescence and confocal microscopy analyses were carried out as described[4]. 9EG7 was diluted in PBS containing 5 mM EGTA and 2 mM $MgCl_2$. Each image represents a single section of 0.7 μm thickness.

**Confocal image analyses.** Picture fields were chosen arbitrarily on the basis of DAPI (4, 6-diamidino-2-phenylindole) staining and images were taken in unsaturated conditions. A minimum of 50–100 cells were analysed per condition per experiment.

For the quantification of HGF-AlexaFluor-555 cellular uptake, the percentage of positive cells or the average red pixels/nuclei (DAPI) were measured using the Zeiss LSM710 Zen software, as indicated in the figure legend.

For double colocalization analysis (for example c-Met-β1-integrin), pixels from each channel were interactively thresholded to remove background pixels using the Zeiss LSM710 Zen software and applied to the whole dataset. The following formula was applied: c-Met-β1 overlapping pixels/total c-Met pixels.

For triple colocalization analysis of c-Met–β1-integrin-endosomal marker, a mask of c-Met–β1 double colocalization was made using the Zeiss LSM710 Zen software. Then triple colocalization of the colocalized c-Met and β1-integrin pixels (= mask) with the intracellular marker pixels was analysed using MetaMorph software. The following formula was used: (c-Met–β1 coloc/endosomal marker = mask)/c-Met–β1 coloc.

Data was further normalized on total c-Met when GFP-tagged constructs were used, when different cells were compared (β1A/β1A-YYFF) or when analysing the accumulation of c-Met–β1-integrin in intracellular compartments upon different time points of HGF stimulation.

Randomization analysis was calculated with the JaCoP plugin[56] for ImageJ (National Institutes of Health, Bethesda). The co-localization index is represented by Pearson's coefficient calculated following Costes randomization (200 cycles) and automatic threshold calculation[57]. The distribution of Pearson's coefficients of randomized images was fitted to a Gaussian distribution, before calculating the P value for differences between the Pearson's coefficient of the actual images, and that of the randomized images. For triple co-localization analysis, two of the three images where combined using the 'Image Calculator' function with operator 'AND' included in ImageJ and the resulting image was tested for co-localization with the third image. All possible permutations were tested and average Pearson's coefficient is provided as result of the triple co-localization.

**Low-light live imaging.** Cells were grown on 35-mm glass-bottom microwell dishes (Matec, Northborough, MA, USA) coated with poly-L-lysine. Time-lapse low-light imaging was acquired on an Axiovert TM 135 microscope (Carl Zeiss) equipped with a 63 × numerical aperture (NA) 1.3 objective lens and an Orca ER CCD camera (Hamamatsu) using Acquisition Manager (Kinetic Imaging). Quicktime movies were constructed from sets of sequential TIFFs using the AQM 2001 Kinetic Acquisition Manager software (Kinetic Imaging, Liverpool, UK).

**Time-lapse confocal.** Cells were cultured on a MatTek dish in phenol red-free DMEM supplemented with 10% FBS. Live cell confocal imaging was performed on LSM710 inverted confocal microscopes equipped with a 63 × 1.4 Plan-Apochromat oil immersion objective (Carl Zeiss). Imaging was performed in an environmental chamber at 37 °C supplemented with 5% $CO_2$. Pictures were acquired every 14 s with a section depth of 1.1 μm.

**Proximity ligation assay.** Cells ($3 \times 10^4$) were cultured and stimulated +/− HGF on coverslips, fixed in 4% PFA and quenched with $NH_4CL$. PLA probing was carried out using PLA probe anti-mouse PLUS and PLA probe anti-goat MINUS kits following manufacturer's protocols (Duolink, Sigma-Aldrich). Samples were incubated in primary antibodies diluted 1:100 in the antibody diluent at room temperature for 50 min. Detection Reagents Orange was used following manufacturer's instructions. Samples were mounted using Duolink *In Situ* Mounting Medium with DAPI, but without air-drying the samples. For the quantification, at least five random fields, across the coverslips based on DAPI staining, corresponding to at least 30 cells per coverslips, were pictured and the number of fluorescence spots/nucleus was quantified using Image J software.

**Flow cytometry.** To determine the level of β1-integrin (activated or pan) at the plasma membrane, cells were trypsinized and washed two times in cold FACS buffer (PBS 2% serum), incubated with antibodies against activated (9EG7, 1/50) or pan (DF7 or MB1.2, 1/100) β1-integrin in cold FACS buffer. After incubation on ice, cells were washed and incubated on ice with PE or APC conjugates. When phospho-ERK1/2 was being analysed, $1 \times 10^6$ cells were fixed in 4% PFA for 10 min at 37 °C and then permeabilized in 90% ice-cold methanol for 30 min on ice. Cells were incubated with a phospho-ERK1/2 antibody (R&D Systems (MAB1018)) at 20 μg ml$^{-1}$ for 1 h. After incubation on ice, cells were washed and incubated on ice with a BD Phosflow PE anti-rabbit secondary antibody (1:5). Flow cytometry data were acquired on a FACS Calibur (Becton Dickinson).

**Biotinylation internalization assay.** Cells were incubated with HGF except the M1268T cells.

On ice, cell surface proteins were labelled with 0:2 mg ml$^{-1}$ sulpho-NHS-SS-biotin in PBS for 45 min. Labelled cells were washed with cold PBS and incubated at 37 °C in culture medium, to allow protein trafficking. At the indicated times, the medium was aspirated and the dishes were transferred to ice and washed with cold

PBS. Biotin was removed from proteins remaining at the cell surface by reduction for 15 min with 180 mM of the membrane-impermeant reducing agent MesNa (sodium 2 mercaptoethane sulphonate, Sigma) in 50 mM Tris and 100 mM NaCl at pH 8.6. MesNa was quenched by the addition of 180 mM iodoacetamide (IAA, Sigma) for 10 min. Cells were lysed. Lysates were passed three times through a 27-gauge needle and clarified by centrifugation (17,000g); equal protein amounts received streptavidin-agarose beads and were agitated at 4 °C for 2 h; beads were collected by centrifugation (7,000g), washed in lysis buffer and proteins were extracted by heating at 95 °C with sample buffer.

In each internalization assay, two controls were carried out. To measure the total c-Met or β1-integrin at the surface, biotinylated cells at 4 °C were lysed without biotin reduction. To verify the efficiency of the surface biotin removal, the biotin reduction and MesNa quenching steps were carried out on cells that had remained on ice (time 0) and lysis was carried out.

Equivalent volumes were analysed in a c-Met or β1-integrin (as relevant) western blotting assay and densitometric analyses were carried out. The percentages of internalized c-Met or β1-integrin were calculated using the following formulae: internalized receptor = (receptor level after incubation at 37 °C) − (receptor level at time 0)/(total surface receptor) × 100.

**Cell adhesion assay.** Cells were detached with trypsin, treated with soybean trypsin inhibitor in serum-free medium, pre-treated with the indicated β1-integrin blocking antibody for 15 min and then seeded onto a well of 24-well plate pre-coated with fibronectin. The cells were incubated for 30 min, rinsed twice with PBS then fixed with 4% paraformaldehyde (PFA) and stained with haematoxylin. Three pictures were taken per condition per experiment with phase contrast microscope and cells counted. At least 50 cells were counted in total per condition per experiment.

**Soft agar assay.** A total of 500 cells in a single-cell suspension were mixed, on ice, in 5 ml of medium with 0.3% agarose. After 20 min, 1 ml of culture medium was added and cells were incubated at 37 °C. Medium was changed daily. For GD25/β1A/β1A-YYFF cells, MRC5 fibroblasts's conditioned media or HGF (14 ng ml$^{-1}$) (as specified in Figure legends) was added, or not, daily, from day 8 and results analysed at day 13. For NIH3T3 WT/M1268T cells, results were analysed at day 6. The wells were pictured on a Zeiss, Stemi SV11 microscope and the total area of the colonies was determined with ImageJ software.

**Tumour growth and metastatic lung assay.** Female nude mice (4-6 weeks old, CD1 Nu/Nu, Charles River UK) were used, in accordance with UK Coordination Committee on Cancer Research guidelines, Home Office regulations and QMUL Ethics boards.

For the tumour growth assay, cells were inoculated subcutaneously in the flank region of nude mice. WT and M1268T ($5 \times 10^5$) c-Met-expressing cells were transfected with control or β1 siRNA 24 h before subcutaneous or tail vein injection. GD25, β1A and β1A-YYFF cells ($5 \times 10^5$) were injected together with $2.5 \times 10^5$ MRC5 fibroblasts. DMSO or PHA-665752 (PHA) was applied topically onto the surface of the skin where GD25, β1A and β1A-YYFF had been injected from day 1 after injection. Tumour volumes were calculated by using the formula: length × width$^2$ × 0.52. When tumours reached 1 cm in length, mice were killed humanely.

For the experimental metastasis assay, $5 \times 10^5$ cells were injected into the tail vein of mice. Ten days later mice were killed and the lungs were removed, weighed and analysed for lung metastasis.

**Zebrafish invasion assay.** The Casper strain (lack pigment) of zebrafish was used. Fish were kept at 28 °C in aquaria with day/night cycles (10-h dark/14-h light periods). Zebrafish embryos were dechorionated and anesthetized with tricaine before injection. Using a manual injector (Picospritzer III), 100 cell tracker' stained cells (50 with Orange CMTMR, for example siRNA control, and 50 with green CMFDA (10 mol l), for example siRNA β1-integrin) were injected together into the yolk sack of 48 h old embryos and embryos maintained at 35 °C. The colour of each cell type was alternated within each experiment to ensure results are not an artefact of the dye. The number of disseminated cells were counted 24 h after injection of the cells, using a Zeiss Axioplan epifluorescence microscope. A minimum of 20 (mean 30) embryos were analysed per condition (for example, WT control siRNA knockdown cells) per experiment. Any embryos showing cells in their body 2 h post injection were removed from the study. When inhibitors were used cells were pre-treated before injection and the inhibitors were added to the water of the zebrafish embryos.

**Statistical analysis.** A two-tailed unpaired Student's t-test was carried out between different conditions. A two-way ANOVA was carried out on the GD25, β1A and β1A-YYFF in vivo tumour growth curves and a Mann–Whitney U test was carried out on the in vivo invasion of A549 cells in zebrafish embryos. Quantitative data of the indicated number of independent experiments ('n =' in figure legends) are expressed as means ± s.e.m.

**Data availability.** The data supporting the findings of this study are available from the corresponding author on request.

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

## Acknowledgements

We thank S. Johansson for his gift of GD25, β1A and β1A-YYFF cells. We thank S. Tooze and O. Florey for discussion and advice regarding the autophagy-related data. R.B.M. was a recipient of a UK Medical Research Council (MRC) studentship, MRC Centenary Award, Barts and The London Charity (472/1711) and Rosetrees Trust (M314), N.K. was a recipient of an MRC studentship (MR/J500409/1), C.J. was a recipient of the Barts and The London Charitable Foundation Scholarship (RAB 05/PJ/07), L.M. was supported by CR-UK, Breast Cancer Now (2008NovPR10) and Rosetrees Trust (M346), A.H. was a recipient of a CR-UK studentship (C236/A11795). P.J.P. was supported by CR-UK. J.I. was supported by grants from the Academy of Finland, ERC Starting grant, Finnish Cancer Organisations and Sigrid Juselius Foundation. S.K. was supported by the MRC (G0501003) and The British Lung Foundation (CAN09-4).

## Author contributions

R.B.M., N.K., C.J and L.M. performed and analysed most of the *in vitro* experiments. A.H., B.A.B., A.J.N., A.M., L.R.M. performed some *in vitro* experiments. X.I. and S.K. constructed the c-Met-GFP construct and generated the TET ON c-Met-GFP cell lines. R.B.M., A.H. and J.H. peformed the *in vivo* mice experiments. R.B.M. performed the *in vivo* zebrafish experiments thanks to C.H.B. equipment, training and expertise. I.R.H. advised on the design of the *in vivo* mice experiments, trained C.J. and R.B.M. and edited the manuscript. C.G. performed the randomization analyses of the confocal pictures. P.J.P contributed to the experimental design, to the interpretation of the results and provided comments on the manuscript. J.I. provided tools and expertise on β1-integrins, performed some experiments and commented on the manuscript. S.K. conceived and directed the project, designed the experiments, performed and analysed some experiments and confocal analyses and wrote the manuscript.

## Additional information

**Competing financial interests:** The authors declare no competing financial interests.

