## [Peer review file · Nature Communications]

Transferred manuscripts:

Reviewers' Comments:

Reviewer #1 (Remarks to the Author)

The authors have constructively revised the manuscript, which now represents a very impressive body of work that makes an important point with many novel aspects.

There are a few minor issues that need to be addressed. The most serious is the assignment of the integrin as a scaffold linking Shc to c-Met. The data show modulation of c-Met/Shc interaction by integrin but it is clearly not essential for co-IP, nor is the biochemistry sufficiently detailed to reach this interpretation. The authors need to be more cautious in their conclusions.

Fig 1b. Where are the images quantified on the right? It's a little confusing to link the images on the left to a different quantification on the right.

Fig 2e,f,g. It is hardly surprising that tumor growth is dependent on integrins. The authors need to be cautious in claiming that this is entirely due to c-met synergy.

Fig 5k-m. Active Erk stays on the endomembranes? This seems unusual and needs to be discussed, with citations of the literature.

Fig 6d. There is a dramatic, perhaps 10 fold or more, upregulation of Shc in the integrin b1 YYFF cells that makes these results hard to interpret. In fact, the change in Shc co-IP with c-met is rather subtle. This result points toward a modulatory role for integrin signaling but is not consistent with an essential adapter or scaffolding function.

Reviewer #2 (Remarks to the Author)

The authors have satisfactorily addressed my concerns from the previous round of review. In particular, they added new data characterizing better the relationship between autophagy and Met/ β 1-integrin co-signaling (showing that Met activation or β 1-integrin levels do not affect basal autophagy, and that autophagy does not contribute to Met degradation). Moreover, additional data using ATG13 depletion indeed argue that canonical autophagy does not underlie the observed phenomenon and generation of ARE. At this stage I do recommend publication of this manuscript. In my view, it documents well an intriguing case of signaling emanating from intracellular compartments due to the cross-talk between two classes of transmembrane receptors.

Reviewer #1 (Remarks to the Author):

The authors have constructively revised the manuscript, which now represents a very impressive body of work that makes an important point with many novel aspects.

We thank the reviewer for acknowledging the improvements of this manuscript and recommending publication.

There are a few minor issues that need to be addressed. The most serious is the assignment of the integrin as a scaffold linking Shc to c-Met. The data show modulation of c-Met/Shc interaction by integrin but it is clearly not essential for co-IP, nor is the biochemistry sufficiently detailed to reach this interpretation. The authors need to be more cautious in their conclusions.

Fig 1b. Where are the images quantified on the right? It's a little confusing to link the images on the left to a different quantification on the right.

We have removed this graph, as it was redundant with the quantification shown below the pictures.

Fig 2e,f,g. It is hardly surprising that tumor growth is dependent on integrins. The authors need to be cautious in claiming that this is entirely due to c-met synergy.

We do not claim that β 1-integrin dependent tumour growth is entirely due to c-Met synergy. As we use tumorigenesis (**Figure 2e**) and invasion (**Figure 2g**) models driven by c-Met activity, our results indicate that the c-Met dependent tumorigenesis and invasion require β 1-integrin, at least in part.

Fig 5k-m. Active Erk stays on the endomembranes? This seems unusual and needs to be discussed, with citations of the literature.

There is also some active ERK1/2 in the cytoplasm but at a lower intensity. The picture has been contrasted to allow a good visualisation of active ERK1/2 on endomembranes. These cells express a constitutive active c-Met-GFP construct, which therefore must dynamically triggers the activation of pools of ERK1/2 on endomembranes. In our view, activated ERK1/2 do not remain on the endomembranes but a pool of active ERK1/2 is always detected on endomembranes. We have modified the text in the result part to reflect this view. The graphs in **Figure 5l and m** represent the percentage of P-ERK1/2 colocalisation with c-Met-LC3B in c-Met-GFP cells knocked down for β 1-integrin and in β 1-YYFF cells compared to the colocalisation, in the respective controls (control knock down or β 1A cells), set as 100%. The legend has been modified to clarify this.

Fig 6d. There is a dramatic, perhaps 10 fold or more, upregulation of Shc in the integrin b1 YYFF cells that makes these results hard to interpret. In fact, the change in Shc co-IP with c-met is rather subtle. This result points toward a

modulatory role for integrin signaling but is not consistent with an essential adapter or scaffolding function.

Although the level of expression of p52^{Shc} is increased in β 1A-YYFF versus β 1A cells, p52^{Shc} co-immunoprecipitation with c-Met is reduced compared to β 1A cells, in basal conditions and upon HGF stimulation. Moreover, there is no significant increase in the co-immunoprecipitation upon HGF versus no HGF in β 1A-YYFF cells while it is significantly increased in β 1A cells. These results, together with results presented **Figure 5Im** are consistent with a scaffolding role of β 1-integrin to our point of view. However, we have modified the text in the manuscript and only suggest this role. We have removed the scaffold function from the abstract and only mention signalling.

Reviewer #2 (Remarks to the Author):

The authors have satisfactorily addressed my concerns from the previous round of review. In particular, they added new data characterizing better the relationship between autophagy and Met/ β 1-integrin co-signaling (showing that Met activation or β 1-integrin levels do not affect basal autophagy, and that autophagy does not contribute to Met degradation). Moreover, additional data using ATG13 depletion indeed argue that canonical autophagy does not underlie the observed phenomenon and generation of ARE. At this stage I do recommend publication of this manuscript. In my view, it documents well an intriguing case of signaling emanating from intracellular compartments due to the cross-talk between two classes of transmembrane receptors.

We thank the reviewer for acknowledging the improvements of this manuscript and recommending publication.